# A Secondary Analysis of Gender Respiratory Features for Ultrasonography Bilateral Diaphragm Thickness, Respiratory Pressures, and Pulmonary Function in Low Back Pain

**Nerea Molina-Hernández [1], David Rodríguez-Sanz [1,*], José López Chicharro [2],**
**Ricardo Becerro-de-Bengoa-Vallejo [1], Marta Elena Losa-Iglesias [3], Davinia Vicente-Campos [4],**
**Daniel Marugán-Rubio [1,5], Samuel Eloy Gutiérrez-Torre [1] and César Calvo-Lobo [1]**

[1] Faculty of Nursing, Physiotherapy and Podiatry, Universidad Complutense de Madrid, 28040 Madrid, Spain; neremoli@ucm.es (N.M.-H.); ribebeva@ucm.es (R.B.-d.-B.-V.); daniel.marugan@lasallecampus.es (D.M.-R.); samgut02@ucm.es (S.E.G.-T.); cescalvo@ucm.es (C.C.-L.)

[2] Grupo FEBIO, Universidad Complutense de Madrid, 28040 Madrid, Spain; jlopezch@ucm.es

[3] Faculty of Health Sciences, Universidad Rey Juan Carlos, 28922 Madrid, Spain; marta.losa@urjc.es

[4] Faculty of Health Sciences, Universidad Francisco de Vitoria, Pozuelo de Alarcón, 28223 Madrid, Spain; davinia.vicente@ufv.es

[5] Centro Superior de Estudios Universitarios La Salle, 28023 Madrid, Spain

\* Correspondence: davidrodriguezsanz@ucm.es

**Abstract:** The aim of the present study was to determine the gender respiratory differences of bilateral diaphragm thickness, respiratory pressures, and pulmonary function in patients with low back pain (LBP). A sample of 90 participants with nonspecific LBP was recruited and matched paired by sex (45 women and 45 men). Respiratory outcomes included bilateral diaphragm thickness by ultrasonography, respiratory muscle strength by maximum inspiratory (MIP) and expiratory (MEP) pressures, and pulmonary function by forced expiratory volume during 1 s ($FEV_1$), forced vital capacity (FVC) and $FEV_1$/FVC spirometry parameters. The comparison of respiratory outcomes presented significant differences ($p < 0.001$), with a large effect size ($d = 1.26$–$1.58$) showing means differences (95% CI) for MIP of $-32.26$ ($-42.99$, $-21.53$) cm $H_2O$, MEP of $-50.66$ ($-64.08$, $-37.25$) cm $H_2O$, $FEV_1$ of $-0.92$ ($-1.18$, $-0.65$) L, and FVC of $-1.00$ ($-1.32$, $-0.69$) L, with lower values for females versus males. Gender-based respiratory differences were presented for maximum respiratory pressures and pulmonary function in patients with nonspecific LBP. Women presented greater inspiratory and expiratory muscle weakness as well as worse lung function, although these differences were not linked to diaphragm thickness during normal breathing.

**Keywords:** diaphragm; disability; low back pain; quality of life; respiration; ultrasonography

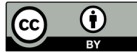

## 1. Introduction

Nonspecific low back pain (LBP) may be defined as a complex disorder without any specific condition or structural reason in the spine to explain the pain in this body region, including factors from various dimensions, such as movement, pain sensitivity, psychological aspects, and work conditions, which may influence both central and peripheral nociceptive processes [1,2]. This condition is considered the most common musculoskeletal disorder, with an estimated cumulative incidence rate of 80% among the population throughout their working lives. This condition may cause chronic disability, reduced quality of life, emotional and social disorders, high healthcare costs, and work absenteeism, as well as a negative influence on functional capacity and other factors linked to biopsychosocial activity such as gender. Female gender has shown an association with a worse prognosis in different aspects in patients suffering from LBP. Therefore, the study of the influence of gender

on LBP is considered of great interest [3–7]. Concretely, LBP presents a point prevalence ranging from 10% to 67%, an annual prevalence ranging from 17% to 94%, and a lifetime prevalence ranging from 33% to 84% [8]. Studies from industrialized countries showed that LBP generated annual costs of up to USD 96 million [9]. In Europe, the direct costs generated by this musculoskeletal pathology exceeded EUR 7000 per person per year [10]. Nonspecific LBP produced an altered function of the trunk stabilizer muscles, including respiratory alterations associated with diaphragm dysfunction [11,12].

Patients suffering from nonspecific LBP presented respiratory alterations [13] which appeared to be associated with a thinner diaphragm muscle [12]. Therefore, respiratory muscle weakness and worse lung function in conjunction with the abnormal position of the diaphragm dome and the altered postural motor control were also shown in these patients [14–16].

Indeed, there is a need to generate novel scientific evidence on the functional effects of diaphragm muscle training during respiratory activity in patients suffering from LBP [17,18], with ventilatory mechanic considered as a key aspect in several biomedical fields [19–21], especially in patients suffering from LBP [22,23]. Despite the key role of the diaphragm muscle being researched for more than 50 years, there is a lack of scientific knowledge about the diaphragm role in LBP disorders [15,24]. The diaphragm muscle has been claimed to be a key stabilizer linked to LBP [12,22]. Nevertheless, gender differences of respiratory features of patients with LBP need to be further studied. Anatomically, the length of the diaphragm is reported to be 9% shorter in women than men, and this length in the zone of apposition with respect to the rib cage is also smaller in women, which is correlated with worse lung function and weaker respiratory muscles [25]. Pathologically, these respiratory features could be exacerbated under nonspecific LBP, showing greater respiratory muscle weakness, worse lung function, and altered diaphragm [14–16].

Regarding recent scientific evidence about diaphragm and respiratory features between men and women, the female diaphragm is greatly fatigue-resistant, which leads to inspiratory muscle metaboreflex attenuation, producing cardiovascular and respiratory consequences [26,27]. In addition, novel evidence suggests that there are sex differences in the change of diaphragm voluntary activation after exercise, although not inspiratory pressure threshold loading, suggesting that diaphragm fatigability may affect exercise performance in humans [28]. Greater diaphragm fatigability was reported in patients with LBP [29]. In spite of the existence of diaphragm thickness and strength differences as well as respiratory features in patients who suffered from LBP with respect to healthy controls [12,22], the influence of gender on these characteristics should be stated. Thus, we hypothesized that gender may influence the thickness and strength diaphragm as well as the respiratory function, in addition to the clinical features of patients suffering from LBP. Therefore, the aim of this study was to determine the gender respiratory differences of bilateral diaphragm thickness, respiratory pressures, and pulmonary function in conjunction with clinical features in patients with LBP.

## 2. Methods

### 2.1. Study Design, Recruitment and Ethical Aspects

A cross-sectional descriptive observational study was carried out from November 2022 to January 2024, performing a secondary analysis of the basal data of the sample of a randomized clinical trial (NCT04582812), according to the Strengthening the Reporting of Observational studies in Epidemiology (STROBE) criteria [30]. Human experimentation ethical aspects and Helsinki Declaration were followed [31]. This research was approved on 18 November 2020, by the ethics committee of the San Carlos Clinical Hospital from Madrid, Spain, with approval code C.I. 20.655-E_BS. Before participating in this study, all participants provided the signed informed consent form. The sample was obtained by a consecutive sampling method from the baseline data of a randomized clinical trial

(NCT04582812), which used a simple randomization method to recruit all participants matched paired by sex in the Complutense University of Madrid.

Furthermore, a patent registry was performed in the Spanish Patent & Trademark Office (application number: U202200045; number of publication: ES1288519; issue date: 30 March 2022), by a utility model for a bilateral thoracic orthosis device which included both left and right holding devices for two ultrasound probes. This device was composed of a thoracic orthosis including 2 bilateral bivalve adapters for 2 holding devices to fix 2 ultrasound probes, allowing total thoracic mobility and including two spaces to incorporate ultrasound gel, permitting the complete visualization of both right and left last intercostal spaces. These holding devices fixed both left and right ultrasound probes on the thoracic orthosis to determine diaphragm bilateral thickness measurements during normal breathing. This thoracic orthosis device minimized diaphragm thickness measurement errors with respect to the probe manual fixation measurement [12,32,33]. This research study was supported and funded by the grant PID2020-117162RA-I00 funded by MICIU/AEI/10.13039/501100011033 corresponding to the Ministry of Science, Innovation and Universities and the State Agency for Investigation from the Spanish Government by the 2020 Call for Innovation, Development and Research ("I + D + i Projects") with the framework of the State Programs for Knowledge Generation and Scientific and Technological Strengthening of the I + D + i System as well as I + D + i oriented to the Challenges of Society.

Descriptive data and outcomes were measured by a physician together with other experienced evaluators using self-reported and interexaminer reliable tools, applying a blinded evaluation for ultrasound images by numerical coding [23].

### 2.2. Sample Size Calculation

A post hoc sample size calculation was performed to justify the achieved power obtained from the baseline data of a randomized clinical trial (NCT04582812) by the difference between 2 independent groups of the version 3.1.9.2 of the G*Power program (G*Power©, University of Dusseldorf; Germany) [34]. Using a general large effect size of Cohen $d = 0.80$ [35], a two-tailed hypothesis, an error probability of $\alpha = 0.05$, and the used sample size of 45 women and 45 men, a power (1-$\beta$ probability error) of 0.96 was obtained.

### 2.3. Study Sample

From the total sample of 96 participants included for eligibility, a final sample of 90 participants with nonspecific LBP was recruited and matched paired by sex (45 women and 45 men). Inclusion criteria comprised patients with a prior medical diagnosis of bilateral nonspecific LBP for more than 6 weeks considering the main pain location referred by the patient between the bi-iliac and subcostal lines with a bilateral positive active straight leg raise test and aged between 18 and 65 years old [12]. The exclusion criteria comprised patients taking painkiller medication, who had received physiotherapy programs in the last 6 months, with a prior medical diagnosis of congenital lumbar disorders, rheumatic, or neuromuscular disorders, body mass index (BMI) greater than 31 kg/m², previous diagnosis of respiratory or neurological pathology, previous surgery and lower limb pathology (including fractures, sprains, or joint instability), skin disorders, inability to follow instructions during the study, pregnancy, and the presence of hyperventilation syndrome assessed by the Nijmegen test greater than 24 points [12,32,36].

### 2.4. Descriptive and Physical Data

Age (y), height (cm), weight (kg), and BMI (kg/cm²) were collected [37]. Using the International Physical Activity Questionnaire (IPAQ), the metabolic equivalent index per minute per week (METs/min/week) was calculated for physical activity determination [38]. Finally, the Nijmegen test was self-reported by all participants in order to show the respiratory distress scores according to the influence of this test on diaphragm activity [12,32,33,39].

*2.5. Respiratory Outcomes*

2.5.1. Bilateral Diaphragm Thickness

Ultrasound measures were carried out by the bilateral thoracic orthosis device, fixing both right and left ultrasound probes to calculate bilateral diaphragm thickness during normal breathing by a randomized assessment order of both hemi-diaphragms. Ultrasound images were coded, saved, and assessed by a blinded examiner by the 2.0 version of the ImageJ software (United States National Institute of Health; Bethesda, Maryland, USA) [12,32,33].

Ultrasound measurements were performed at transcostal location for both right and left hemi-diaphragm thickness (cm) at maximum inspiration ($T^{ins}$) and expiration ($T^{exp}$), and calculating their differences ($T^{ins-exp}$) during normal breathing. Two high-quality ultrasound devices were used to determine all ultrasound measurements and images (Ecube-i7; Alpinion from Medical Systems; Seoul, Korea). All images were taken using 2 linear probes (L3_12T-types; 34 mm field of view; with 128 elements), with a frequency from 8 MHz to 12.0 MHz and a footprint of 45 mm. These measurements were performed at supine position by B-mode ultrasound imaging with a prefixed preset including 3 cm deep, 12 MHz frequency, 64 points gain, 64 points dynamic range, and 1 focus located at 2 cm depth [12,17,32,33]. All ultrasound images were taken in grayscale and converted in the format of Digital Imaging and Communications in Medicine (DICOM), calibrated by the 2.0 v-ImageJ software (U.S.-National Institutes of Health; Bethesda, MD, USA) to determine the thickness of both hemi-diaphragms [12,32,33]. Furthermore, the linear probes were placed perpendicular to the last intercostal spaces following the mid-axillary line from the lower edge of the 11th rib to the upper edge of the 12th rib of the thorax region, permitting a correct diaphragm visualization under connective tissue of intercostal muscles during normal breathing activity (Figure 1). A total of 3 repeated measures were carried out to measure both right and left hemi-diaphragms thickness at $T^{ins}$, $T^{exp}$, and $T^{ins-exp}$, using 3 images for each parameter, which showed a reduction in errors of measurement [32]. Both hemi-diaphragms' thickness measurements were carried out by placing each electronic caliper inside of the upper and lower hyper-echogenic lines from the connective tissue around the diaphragm muscle, locating the thickness measurements at the center of the intercostal space. A total of 3 repeated measurements were applied to calculate the final mean. Both right and left probes' fixation was performed by the bilateral orthotic device according to the manual measurement errors reductions, and separate unilateral measurements were carried out according to the better reliability parameters, showing excellent reliability to determine ultrasound thickness measures of the diaphragm muscle during normal breathing, with intraclass correlation coefficients (ICC) of 0.852–0.996, standard errors of measurement (SEM) of 0.0002–0.054 cm, and minimum detectable changes (MDC) of 0.002–0.072 cm, avoiding systematic measurement errors [32,33].

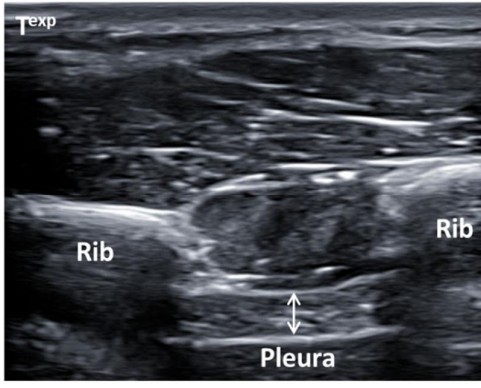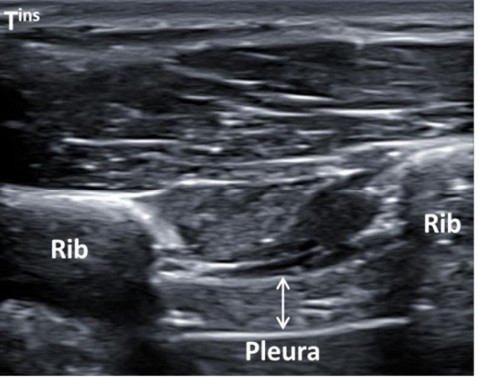

**Figure 1.** Ultrasonography images by B-mode of the diaphragm thickness that illustrated the last intercostal spaces coinciding with the mid-axillary line between the lower edge of the 11th rib and

the upper edge of the 11th rib of the thorax region. ($T^{exp}$) Diaphragm thickness measurements were determined by the white arrow at maximum expiration ($T^{exp}$) during normal breathing placed at the left side of this figure. ($T^{ins}$) Diaphragm thickness measurements were determined by the white arrow at maximum inspiration ($T^{ins}$) during normal breathing placed at the right side of this figure.

### 2.5.2. Respiratory Pressures

The strength of the inspiratory and expiratory muscles was measured by maximum expiratory (MEP) and inspiratory (MIP) pressures, respectively, by the RP Check device (MD-Diagnostics Ltd.; Chathman, United Kingdom), considering the residual volume, following the protocol recommended by the American Thoracic Society as well as the European Respiratory Society [40,41]. These maximum respiratory pressures were determined in cm $H_2O$ in order to compare both groups under absolute values. This protocol was repeated at least 3 times or up to 2 reproducible efforts (within 5% for each other). An interval of 1 minute was added between these measures in order to avoid the respiratory muscular fatigue in this short term. The greatest value of 2 reproducible measures was used for data analysis [22]. This protocol showed excellent interexaminer reliability, presenting an ICC that varied from 0914 to 0.925 [42].

### 2.5.3. Pulmonary Function

Spirometry respiratory parameters were analyzed to determine airway airflow restrictions by the Datospir-600 Touch device (SIBELMED e-20; Barcelona, Spain). Spirometry respiratory parameters included the forced expiratory volume during 1 second ($FEV_1$; measured in L), the forced vital capacity (FVC; measured in L), and the $FEV_1$/FVC coefficient (measured in %). These data may be considered as the most important respiratory parameters to show airway disturbances at physiological level. These values showed that lung functions were prior correlated with chest wall expansion, determining $r = 0.74$. A good reliability was shown for these parameters, presenting an ICC that varied from 0.786 to 0.929 [43].

### *2.6. Clinical Outcomes*

### 2.6.1. Pain Intensity

Pain intensity measurements were determined using the Visual Analog Scale (VAS) by the self-reported pain intensity average during the last week at rest according to the methodology used in prior similar studies [12,23,32]. This scale was composed of a 100 mm horizontal line where participants marked their pain intensity from "no pain" at the left side to the "worst pain imaginable" at the right side. This scale presented adequate reliability as validity, presenting an ICC of 0.88 to determine pain intensity in the last week with a correlation of $r = 0.74$ with other scales of pain intensity [5,44].

### 2.6.2. Pressure Pain Threshold

Pressure pain threshold (PPT) measurements were determined from 0 to 10 kg/cm$^2$ by a mechanical algometer device (Wagner-Instruments, Greenwich, CT, USA). This tool showed ICC = 0.91, variation coefficient = 10.3%, SEM = 0.19 kg/cm$^2$, and MDC = 0.54 kg/cm$^2$. These values presented adequate reliability, sensitivity, and reproducibility to assess the PPT placed at the center of the paravertebral spinal muscles bilaterally by perpendicular location with respect to the L3 spinous process. These measurements were carried out manually using a gradual protocol up to the point where the patient felt an initial pain sensation. This protocol was performed 3 times at the same location using an interval from 30 to 60 seconds and calculating the mean of 3 repeated measures [45,46].

### 2.6.3. Disability

This clinical outcome was measured using the Spanish Roland-Morris Disability Questionnaire (RMDQ), which has been widely used to self-report the disability linked to LBP following prior similar studies [12,23,32], showing adequate validity and reliability

with ICC = 0.87, and using 24 items which assess the daily life limitations due to LBP, from 0 ("no disability") to 24 points ("maximum disability") [12,47].

### 2.6.4. Quality of Life

Quality of life was determined using the Short-Form 12-items (SF-12) health questionnaire to estate the health-related quality of life measurements by optimal normalized values to determine mental and physical health domains and the total scores. This tool presented adequate psychometric properties, validity, and reliability, showing Cronbach $\alpha$ from 0.78 to 0.85 [48].

### *2.7. Statistical Analyses*

The statistical analyses were performed by the 24.0 version of the Statistical Package of Social Sciences (SPSS) (IBM; Armonk, NY, USA; IBM-Corp), applying $\alpha$ error = 0.05 and *p*-value < 0.05 considered as significant using confidence interval (CI) = 95%. Statistical analyses were carried out to compare 2 groups. The Kolmogorov–Smirnov test was applied to determine the distribution normality as this statistical test is recommended in health science fields for sample sizes with more than 30 subjects per group [49]. Parametric and nonparametric data are described using the mean ± standard deviation (SD) and upper and lower limits of the 95% C, as well as the median ± interquartile range (IR) and maximum and minimum values range. First, the Student's *t*-test was applied to compare differences between both groups considering parametric data for independent samples by the Levene test considering equality of variances. Second, differences between both groups regarding nonparametric data were assessed using the Mann–Whitney *U*-test for independent samples. Furthermore, effect size of the respiratory and clinical outcomes differences between both groups was analyzed by Cohen's *d* using the formula $d = 2t/\sqrt{gdl}$, and categorizing the effect size as very small (*d* < 0.20), small (*d* = 0.20–0.49), medium (*d* = 0.50–0.79), and large (*d* ≥ 0.80) [12,50].

Lastly, the influence of the descriptive and physical data and gender on the respiratory outcomes that presented statistically significant differences between female and male patients with nonspecific LBP was predicted by multivariate linear regression analyses. Four linear regression analyses were carried out for each statistically significant respiratory outcome using the "stepwise selection" method. Each regression coefficient value ($R^2$) was calculated to establish the adjustment quality according to the required number of participants per variable [51]. Therefore, age (years), height (cm), weight (kg), BMI (kg/cm$^2$), IPAQ and Nijmegen scores, and gender (females = 1; males = 2) were included as independent variables. Each aforementioned statistically significant respiratory outcome measurement was included as a dependent variable. Indeed, the pre-established *F* probabilities were between $P_{in}$ of 0.05 and $P_{out}$ of 0.10, respectively [12,23].

## 3. Results

### *3.1. Descriptive and Physical Data*

A final sample of 90 patients with nonspecific LBP were matched, paired into female (n = 45) and male (n = 45) participants. Both groups did not show statistically significant differences (*p* > 0.05) for age or physical activity according to IPAQ score. In addition to the mean difference (95% CI) of height of −15.22 (−17.71, 12.73) cm, weight of −17.19 (−21.56, −12.82) kg, and BMI of −1.42 (−2.73, −0.11) presented by female versus male participants, there were also statistically significant differences (*p* < 0.05) for greater respiratory distress scores with a mean difference (95% CI) of Nijmegen test of 2.93 (0.46, 5.40) points for women with respect to men. These results are shown in Table 1.

**Table 1.** Descriptive data for male and female patients with nonspecific LBP.

| Descriptive and Physical Data (n = 90) | Female LBP Patients (n = 45) | | Male LBP Patients (n = 45) | | *p*-Value |
|---|---|---|---|---|---|
| | Mean ± SD (95% CI) | Median ± IR (Range) | Mean ± SD (95% CI) | Median ± IR (Range) | |
| Age (y) | 49.15 ± 8.89 (46.48–51.82) | 50.00 ± 11.00 (24.00–62.00) | 45.75 ± 13.05 (41.83–49.67) | 50.00 ± 14.50 (18.00–64.00) | 0.343 [†] |
| Height (cm) | 162.76 ± 4.73 (161.33–164.18) | 163.00 ± 6.00 (152.00–172.00) | 177.98 ± 6.92 (175.90–180.06) | 178.00 ± 10.00 (162.00–195.00) | **<0.001** [†] |
| Weight (kg) | 64.64 ± 9.68 (61.77–67.52) | 63.00 ± 17.50 (49.00–80.00) | 81.84 ± 11.22 (78.47–85.21) | 82.00 ± 16.50 (49.00–105.00) | **<0.001** [*] |
| BMI (kg/m$^2$) | 24.37 ± 3.28 (23.38–25.36) | 26.61 ± 5.33 (19.38–30.48) | 25.80 ± 2.95 (24.91–26.68) | 26.17 ± 3.61 (16.76–30.67) | **0.033** [*] |
| IPAQ (METs/min/week) | 1619.34 ± 1136.88 (1277.78–1960.90) | 1404.34 ± 1455.00 (160.00–5544.00) | 2416.77 ± 1888.78 (1849.31–2984.22) | 1584.00 ± 2385.00 (184.80–8586.00) | 0.062 [†] |
| Nijmegen (scores) | 13.95 ± 5.57 (12.28–15.62) | 14.00 ± 7.50 (0.00–22.00) | 11.02 ± 6.20 (9.15–12.88) | 11.00 ± 10.00 (2.00–23.00) | **0.020** [*] |

Abbreviations*:* BMI, body mass index; CI, confidence interval; IPAQ, International Physical Activity Questionnaire; IR, interquartile range; LBP, low back pain; METs/min/week, metabolic equivalent index per minute per week; *p* < 0.05 was significant (**in bold**) considering a 95% CI. [*] Student's *t*-test for independent samples was analyzed. [†] Mann–Whitney-*U* test for independent samples was analyzed.

### 3.2. Respiratory Outcomes

The comparison of respiratory outcomes between both groups presented statistically significant differences (*p* <.001) with a large effect size (*d* = 1.26–1.58) showing means differences (95% CI) for MIP of −32.26 (−42.99, −21.53) cm $H_2O$, MEP of −50.66 (−64.08, −37.25) cm $H_2O$, $FEV_1$ of −0.92 (−1.18, −0.65) L, and FVC of −1.00 (−1.32, −0.69) L, with lower values for female versus male patients with nonspecific LBP. Nevertheless, there were not statistically significant differences (*p* > 0.05) with effect sizes from very small to small (*d* = 0.00–0.33) for bilateral diaphragm thickness at $T^{ins}$, $T^{exp}$, and $T^{ins-exp}$ during normal breathing between both female and male participants. These findings are shown in Table 2.

**Table 2.** Comparisons for respiratory outcomes between female and male patients with nonspecific LBP.

| Respiratory Outcome Differences (n = 90) | Female LBP Patients (n = 45) | | Male LBP Patients (n = 45) | | Cohen *d* | *p*-Value |
|---|---|---|---|---|---|---|
| | Mean ± SD (95% CI) | Median ± IR (Range) | Mean ± SD (95% CI) | Median ± IR (Range) | | |
| Right diaphragm thickness at $T^{ins}$ (cm) | 0.20 ± 0.06 (0.18–0.22) | 0.19 ± 0.00 (0.00–0.36) | 0.22 ± 0.06 (0.20–0.24) | 0.22 ± 0.10 (0.12–0.38) | 0.33 | 0.197 [*] |
| Right diaphragm thickness at $T^{exp}$ (cm) | 0.18 ± 0.04 (0.16–0.19) | 0.17 ± 0.05 (0.11–0.35) | 0.19 ± 0.05 (0.17–0.21) | 0.18 ± 0.07 (0.10–0.34) | 0.22 | 0.253 [†] |
| Right diaphragm thickness at $T^{ins-exp}$ (cm) | 0.02± 0.05 (0.00–0.04) | 0.02 ± 0.04 (−0.17–0.18) | 0.02 ± 0.03 (0.01–0.03) | 0.02 ± 0.04 (−0.03–0.12) | 0.00 | 0.651 [†] |
| Left diaphragm thickness at $T^{ins}$ (cm) | 0.23 ± 0.07 (0.20–0.25) | 0.21 ± 0.08 (0.11–0.48) | 0.21 ± 0.05 (0.19–0.22) | 0.21 ± 0.07 (0.12–0.38) | 0.32 | 0.264 [†] |
| Left diaphragm thickness (cm) at $T^{exp}$ | 0.19 ± 0.05 (0.19–0.21) | 0.19 ± 0.09 (0.10–0.33) | 0.18 ± 0.05 (0.16–0.19) | 0.17 ± 0.06 (0.08–0.31) | 0.20 | 0.285 [*] |
| Left diaphragm thickness at $T^{ins-exp}$ (cm) | 0.03 ± 0.04 (0.02–0.04) | 0.02 ± 0.05 (−0.05–0.16) | 0.02 ± 0.03 (0.01–0.03) | 0.02 ± 0.03 (−0.04–0.18) | 0.28 | 0.990 [†] |
| MIP (cm $H_2O$) | 53.42 ± 19.17 (47.66–59.18) | 48.67 ± 25.33 (17.67–100.33) | 85.68 ± 30.62 (76.48–94.88) | 84.67 ± 46.00 (33.67–153.33) | 1.26 | **<0.001** [†] |
| MEP | 76.88 ± 24.79 | 77.33 ± 27.50 | 127.55 ± 37.77 | 128.33 ± 54.17 | 1.58 | **<0.001** [†] |

| (cm H$_2$O) | (69.43–84.33) | (27.73–150.33) | (116.20–138.90) | (52.67–185.67) | | |
|---|---|---|---|---|---|---|
| FEV$_1$ | 2.51 ± 0.56 | 2.51 ± 0.64 | 3.44 ± 0.69 | 3.57 ± 0.88 | 1.48 | **<0.001 \*** |
| (L) | (2.34–2.68) | (1.37–3.69) | (3.23–3.64) | (1.87–4.93) | | |
| FVC | 2.68 ± 0.63 | 2.72 ± 0.85 | 3.69 ± 0.84 | 3.80 ± 1.07 | 1.36 | **<0.001 \*** |
| (L) | (2.49–2.87) | (1.39–3.96) | (3.43–3.94) | (1.87–6.03) | | |
| FEV$_1$/FVC | 93.70 ± 5.70 | 95.47 ± 9.41 | 93.95 ± 6.59 | 96.58 ± 9.80 | 0.04 | 0.264 † |
| (%) | (91.99–95.41) | (78.90–99.83) | (91.97–95.93) | (75.80–99.94) | | |

*Abbreviations:* CI, confidence interval; FEV$_1$, forced expiratory volume for 1 s; FVC, forced vital capacity; IR, interquartile range; LBP, low back pain; MEP, maximum expiratory pressure; MIP, maximum inspiratory pressure; T$^{ins}$, maximum inspiration time; T$^{exp}$, maximum expiration time. $p < 0.05$ was significant considering a 95% CI (**in bold**). \* Student's *t*-test for independent samples was applied. † Mann–Whitney *U*-test for independent samples was applied.

### 3.3. Clinical Outcomes

Lastly, the comparison of clinical outcomes between both groups displayed significant differences ($p < 0.05$), with an effect size from medium to large ($d = 0.50$–1.12) presenting means differences (95% CI) for right and left paraspinal PPT of −1.81(−2.48, −1.13) kg/cm$^2$ and −1.77 (−2.48, −1.06) kg/cm$^2$, respectively, RMDQ score of 1.55 (0.30, 2.80) points, and the SF-12 score of physical health domain of −8.24 (−15.03, −1.45) points indicating lower bilateral paraspinal PPT, greater disability, and worse quality of life related to physical health for female versus male participants with nonspecific LBP. However, there were not significant differences ($p > 0.05$) with effect sizes from very small to small ($d = 0.04$–0.43) VAS as well as mental health domain and total scores of the SF-12 questionnaire between both groups. These findings were shown in Table 3.

**Table 3.** Comparisons for clinical outcomes between female and male patients with nonspecific LBP.

| Clinical Outcome Differences (n = 90) | Female LBP Patients (n = 45) | | Male LBP Patients (n = 45) | | Cohen *d* | *p*-Value |
|---|---|---|---|---|---|---|
| | Mean ± SD (95% CI) | Median ± IR (Range) | Mean ± SD (95% CI) | Median ± IR (Range) | | |
| VAS | 4.91 ± 1.96 | 4.80 ± 3.10 | 4.82 ± 1.74 | 5.00 ± 2.40 | 0.04 | 0.808 \* |
| (scores) | (4.32–5.50) | (1.50–8.90) | (4.29–5.34) | (0.70–8.80) | | |
| Paraspinal right PPT | 3.72 ± 1.35 | 3.80 ± 2.20 | 5.53 ± 1.84 | 5.67 ± 2.65 | 1.12 | **<0.001 \*** |
| (kg/cm$^2$) | (3.31–4.13) | (1.40–6.33) | (4.98–6.08) | (1.90–10.00) | | |
| Paraspinal left PPT | 3.70 ± 1.29 | 3.67 ± 1.84 | 5.48 ± 2.00 | 5.47 ± 2.74 | 1.05 | **<0.001 †** |
| (kg/cm$^2$) | (3.31–4.09) | (1.30–6.80) | (4.87–6.08) | (1.50–9.93) | | |
| RMDQ | 5.06 ± 3.15 | 4.00 ± 3.50 | 3.51 ± 2.77 | 3.00 ± 2.50 | 0.52 | **0.009 †** |
| (scores) | (4.11–6.01) | (1.00–12.00) | (2.67–3.34) | (0.00–11.00) | | |
| SF-12 Physical health | 62.55 ± 18.06 | 64.00 ± 25.50 | 70.80 ± 14.12 | 71.00 ± 18.50 | 0.50 | **0.023 †** |
| (optimal normalized values) | (57.12–67.98) | (0.00–86.00) | (66.55–75.04) | (36.00–93.00) | | |
| SF-12 Mental health | 64.20 ± 15.57 | 67.00 ± 19.00 | 67.86 ± 12.02 | 67.00 ± 14.00 | 0.26 | 0.280 † |
| (optimal normalized values) | (59.52–68.87) | (29.00–95.00) | (64.25–71.47) | (38.00–90.00) | | |
| SF-12 Total score | 63.48 ± 15.31 | 66.00 ± 20.00 | 69.02 ± 9.71 | 71.00 ± 12.50 | 0.43 | 0.064 † |
| (optimal normalized values) | (58.88–68.09) | (17.00–91.00) | (66.10–71.94) | (43.00–89.00) | | |

*Abbreviations:* CI, confidence interval; IR, interquartile range; LBP, low back pain; MEP, maximum expiratory pressures; MIP, maximum inspiratory pressures; PPT, pressure pain threshold; RMDQ, Roland-Morris Disability Questionnaire; SF-12, Short-Form 12-items health questionnaire; VAS, Visual Analog Scale. $p < 0.05$ was significant considering a 95% CI (**in bold**). \* Student's *t*-test for independent samples was applied. † Mann–Whitney *U*-test for independent samples was applied.

*3.4. Multivariate Linear Regression Analyses*

The first linear regression model ($R^2$ = 0.290) predicted higher MIP values based on the male gender ($R^2$ = 0.290; β = 32.226; $F_{(1,88)}$ = 35.893; $p < 0.001$). Next, the second linear regression model ($R^2$ = 0.391) predicted greater MEP values based on the male gender ($R^2$ = 0.391; β = 50.667; $F_{(1,88)}$ = 56.572; $p < 0001$). Also, the third linear regression model ($R^2$ = 0.351) predicted higher $FEV_1$ based on the male gender ($R^2$ = 0.351; β = 0.924; $F_{(1,88)}$ = 47.624; $p < 0.001$). Finally, the last linear regression model ($R^2$ = 0.355) predicted greater FVC values based on the male gender ($R^2$ = 0.351; β = 0.924; $F_{(1,88)}$ = 47.624; $p < 0.001$) and higher IPAQ scores ($R^2$ = 0.039; β = 1.098; $F_{(1,87)}$ = 5.295; $p = 0.024$). The rest of the independent variables, such as height, weight, BMI, and Nijmegen scores, were excluded from the prediction models according to the pre-established F probabilities for $P_{in}$ of 0.05 and $P_{out}$ of 0.10, and, thus, did not predict or influence these respiratory outcomes.

## 4. Discussion

The present study provides novel evidence showing clinical and respiratory differences between females and males suffering from nonspecific LBP. Up to date, prior studies were mainly focused on the existence of clinical and respiratory differences between participants with and without nonspecific LBP [5,12,24,29,52]. To our knowledge, our descriptive study claims the importance of gender on respiratory outcomes and reinforces its key role in clinical outcomes in line with recent studies [53–56], showing that especially female patients with nonspecific LBP displayed worse clinical and respiratory findings.

Regarding the respiratory outcomes, lower inspiratory and expiratory strength measured by maximum respiratory pressures were presented in female with respect to male LBP patients. These findings may be related to greater respiratory muscles fatigability in women versus men. Indeed, this fatigability was increased in patients suffering from LBP [26,27]. In addition, $FEV_1$ and FVC spirometry parameters suggested a worse pulmonary function in women versus men who suffered from LBP. In addition to the respiratory muscles weakness in patients who suffered from nonspecific LBP of our study, this worse lung function was reported as a key spirometry parameter in patients with LBP [57], which may be improved after visual respiratory biofeedback reeducation [32,33]. Nevertheless, women and men presented a similar bilateral diaphragm thickness during normal breathing. Possibly, this issue may be due to both female and male patients with nonspecific LBP presenting a diaphragm thickness reduction mainly secondary to the presence of LBP [12].

Mainly, female gender predicted lower MIP, MEP, $FEV_1$, and FVC values in patients who suffered from nonspecific LBP according to our multivariate linear regression models ($R^2$ = 0.290–0.391). Previously, men showed an increased diaphragm thickness change compared with healthy women adults, which was positively correlated with lung function parameters and respiratory muscle strength [58]. Our findings were in accordance with spirometry parameters and respiratory pressures differences by gender and in contrast with the differences reported in diaphragm thickness in healthy participants [25,59], suggesting the absence of diaphragm thickness differences by gender in patients with LBP. Possibly, the inhibition of the diaphragm function presented in patients with nonspecific LBP suggesting an altered breathing pattern during lumbopelvic motor control may explain the dilution of the diaphragm thickness differences between men and women under this condition [60].

According to our clinical outcome findings, increased mechanosensitivity secondary to lower PPT in the erector spinae muscles, greater disability, and worse quality of life related to physical health were presented in female versus male patients with nonspecific LBP. Similar results were reported in prior studies, highlighting the importance of gender on these clinical outcomes in LBP patients [53–56]. Pain intensity did not show differences between both women and men with nonspecific LBP. Both groups presented moderate pain intensity under LBP condition and in spite of pain intensity previously associated

with gender in chronic spinal pain, only pain catastrophizing directly influenced the pain intensity [54].

### 4.1. Limitations

Using the secondary analysis of the baseline sample data of a randomized clinical trial (NCT04582812), the present cross-sectional study tried to describe the respiratory and clinical differences between female and male patients suffering from nonspecific LBP, although the authors recognize that the comparison of cases and control participants matched paired by gender could improve the specific comparisons among male and female participants with and without nonspecific LBP [12]. Despite sex and gender terms being used interchangeably in our study according to most research studies, the importance of sex-based and gender-based recommendations for future LBP clinical practice guidelines needs to be considered [55]. In addition, maximum respiratory pressures were measured in cm $H_2O$ to compare both women and men under absolute values following recommendation to avoid bias. However, the use of non-normalized values could lead to misinterpretation [40,41]. Pain intensity was determining, indicating that patients self-reported their LBP intensity average during the last week at rest. Authors acknowledge that the moderate pain intensity mean presented in both women and men could have been confounded by pain intensity during daily life activities with respect to pain intensity at rest, and this issue should be considered in future studies differentiating between pain intensity at rest and during physical activity [5,44]. Furthermore, different physical activity levels should be separately analyzed due to the different possible ventilator patterns presented under different training types [61]. Lastly, our study considered nonspecific LBP criteria according to a prior study's methodology [12], but future studies should include questionnaires or medical imaging techniques to confirm this diagnosis in a more accurate manner in addition to including more descriptive information such as social or working status of the study participants [55].

### 4.2. Future Recommendations

This study provides novel insights into the respiratory outcomes in conjunction with the known clinical outcomes between females and males suffering from nonspecific LBP [5,12,15,24,29,52]. Thus, our study reinforces the importance of analyses stratified by sex under respiratory re-education interventions in patients with nonspecific LBP [23,32], due to female patients with nonspecific LBP presenting worse clinical and respiratory findings.

As clinical recommendations and take-home messages, rehabilitation protocols should pay special attention to improving diaphragm strength, lung function, and clinical outcomes in women with nonspecific LBP, according to our current findings suggesting worse results compared with men, in line with prior studies in healthy adults [25,58,59]. Next, the motor control diaphragm re-education during normal breathing should be applied with the same emphasis in both women and men with LBP, in line with our study findings reporting no diaphragm thickness differences by gender and the altered motor control reported in patients suffering from this condition [60]. Finally, this study only describes baseline data differences by gender, but clinical trials should determine the influence of gender on the respiratory and clinical outcomes during breathing re-education interventions in patients with nonspecific LBP [22,23].

### 5. Conclusions

Gender-based respiratory differences were presented for maximum respiratory pressures and pulmonary function in patients with nonspecific LBP. Women presented greater inspiratory and expiratory muscle weakness as well as worse lung function, although these differences were not linked to diaphragm thickness during normal breathing. These worse respiratory outcomes were mainly predicted by female gender. In addition,

increased mechanosensitivity, greater disability, and worse quality of life related to physical health were presented in female versus male patients with nonspecific LBP.

## 6. Patents

A patent registry was performed in the Spanish Patent & Trademark Office (application number: U202200045; number of publication: ES1288519; issue date: 30 March 2022), by a utility model for a bilateral thoracic orthosis device which included both left and right holding devices for two ultrasound probes.

**Author Contributions:** Conceptualization, D.R.-S., J.L.C., R.B.-d.-B.-V. and C.C.-L.; data curation, D.R.-S., J.L.C., R.B.-d.-B.-V., D.V.-C., D.M.-R., S.E.G.-T. and C.C.-L.; formal analysis, N.M.-H., D.R.-S., J.L.C., R.B.-d.-B.-V., M.E.L.-I. and C.C.-L.; funding acquisition, D.R.-S., J.L.C., R.B.-d.-B.-V. and C.C.-L.; investigation, N.M.-H., D.R.-S., J.L.C., R.B.-d.-B.-V., D.V.-C., D.M.-R., S.E.G.-T. and C.C.-L.; methodology, N.M.-H., D.R.-S., J.L.C., R.B.-d.-B.-V., M.E.L.-I., and C.C.-L.; project administration, D.R.-S. and C.C.-L.; resources, D.R.-S., J.L.C., R.B.-d.-B.-V. and C.C.-L.; software, D.R.-S., J.L.C., R.B.-d.-B.-V. and C.C.-L.; supervision, D.R.-S. and C.C.-L.; validation, N.M.-H., D.R.-S., J.L.C., R.B.-d.-B.-V., D.V.-C., D.M.-R., S.E.G.-T. and C.C.-L.; visualization, D.R.-S., J.L.C., R.B.-d.-B.-V., M.E.L.-I. and C.C.-L.; writing—original draft, D.R.-S., J.L.C., R.B.-d.-B.-V. and C.C.-L.; writing—review and editing, N.M.-H., D.R.-S., J.L.C., R.B.-d.-B.-V., M.E.L.-I., D.V.-C., D.M.-R., S.E.G.-T. and C.C.-L. All authors have read and agreed to the published version of the manuscript.

**Funding:** Grant PID2020-117162RA-I00 funded by MICIU/AEI/10.13039/501100011033 corresponding to the Ministry of Science, Innovation and Universities and the State Agency for Investigation from the Spanish Government by the 2020 Call for Innovation, Development and Research ("I + D + i Projects") with the framework of the State Programs for Knowledge Generation and Scientific and Technological Strengthening of the I + D + i System as well as I + D + i oriented to the Challenges of Society.

**Institutional Review Board Statement:** The study was approved by the ethics committee of the Hospital Clínico San Carlos (Madrid, Spain) with 20.655-E_BS approval code. All participants included in the study received an information sheet and signed an informed consent.

**Informed Consent Statement:** All participants included in the study received an information sheet and signed an informed consent.

**Data Availability Statement:** Raw data will be available as a supplementary file with the future clinical trial publication (NCT04582812) due to this study being a secondary analysis of the baseline data of this clinical trial.

**Acknowledgments:** Authors acknowledge support from grant PID2020-117162RA-I00 funded by MICIU/AEI/10.13039/501100011033 from the Ministry of Science and Research as well as the State Agency for Investigation of the Spanish Government under the 2020 Call for Innovation, Development and Research ("I + D + i Projects") within the framework of the State Programs for Knowledge Generation and Scientific and Technological Strengthening of the I + D + i System and I + D + i oriented to the Challenges of Society (grant number PID2020-117162RA-I00).

**Conflicts of Interest:** The authors declare competing financial interests due to a patent registry being carried out as a utility model for the bilateral thoracic orthosis including both right and left holding devices for 2 ultrasound probes in the Spanish Patent and Trademark Office (number of application: U202200045; publication number: ES1288519; issue date: 30 March 2022).

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
