# Peer review of "A Secondary Analysis of Gender Respiratory Features for Ultrasonography Bilateral Diaphragm Thickness, Respiratory Pressures, and Pulmonary Function in Low Back Pain"

_tomography, doi:10.3390/tomography10060067_

Round 1
Reviewer 1 Report
Comments and Suggestions for Authors
Dear Authors,
The manuscript is interesting, but I argue about the clinical meaning of the results as you did not compare to controls. Please, see my comments below.
- Posteriori means post hoc sample size calculation? If yes, please use the actual description.
- If you are performing a post hoc sample size analysis, in fact, you are performing a sample power analysis. You already have the effect size, obtained from your actual sample. Please, clarify and change in your current draft.
- "Inclusion criteria comprised patients with a medical diagnosis of bilateral 110 non-specific LBP" - How those patients were classified as non-specifi LBP? Were they assessed using a questionnaire (Start-Back, Oswestry, Rolland Moris, etc)? Did you confirm the diagnostic? How the mechanic cause was excluded?
- Non-parametric data must be presented as median, minimum and maximum values.
- Please, explain the clinical value of comparing male vs. female, as the anthropometrics are distinct. Can those characteristics act as co-variables? If yes, their interference should be accounted for your results?
- The fundamental question about the results is: Compared to healthy controls, would those patients show any marked differences? Why didn't you compare the results to controls?
- The discussion is very superficial. The authors must put more effort to explain the results, contrast them to the literature, show and discuss the clinical significance of the current findings, and report the highlights of the present study (take-home messages).
- I strongly recommend to deposit the raw data in a open repository.
Comments on the Quality of English LanguagePlease, check the entire text. Some typos were detected.
Author Response
DATE: 05/20/2024
To: Mr. Chauncey Jiang
Section Managing Editor
Tomography
ID: Manuscript ID: tomography-2999292; Title: “A Secondary Analysis of Gender Respiratory Features for Ultrasonography Bilateral Diaphragm Thickness, Respiratory Pressures and Pulmonary Function in Low Back Pain”
Dear Mr. Chauncey Jiang
Section Managing Editor
Tomography:
We appreciate the opportunity to revise our R1 manuscript ID tomography-2999292, updated and entitled "A Secondary Analysis of Gender Respiratory Features for Ultrasonography Bilateral Diaphragm Thickness, Respiratory Pressures and Pulmonary Function in Low Back Pain” to be considered for publication in the original article manuscript category of the Special Issue “Novel Imaging Advances in Physiotherapy” of Tomography. We trust that you will find the current version informative to your readership and acceptable for publication.
Thanks for your review commentaries in order to improve the quality of the manuscript. A deep and substantial modification has been carried out according to your suggestions.
Modifications to the manuscript text are denoted by line numbers and yellow highlighted in the marked-up copy of our manuscript. Please find our responses to each reviewers and editor comment below.
Sincerely, the authors
RESPONSES TO EDITOR'S COMMENTS:
Thanks for your commentaries in order to improve the quality of the manuscript. A deep and substantial modification has been carried out according to your suggestions.
- Comments
“Your manuscript has been reviewed by experts in the field and we request that you make major revisions before it is processed further. Please revise your manuscript according to the reviewers' comments and upload the revised file within 10 days. Please click on the "Peer Review Reports" below to find the reviewers' comments and the version of your manuscript to be used for your revisions”:
- Thanks for your considerations and review process in order to improve the quality of this manuscript. Please, see complete review in response to each reviewer´s suggestion. Thanks.
RESPONSES TO REVIEWERS' COMMENTS:
Reviewer 1:
Thanks for your commentaries in order to improve the quality of the manuscript. A deep and substantial modification has been carried out according to your suggestions.
Comment 0: “Dear Authors, The manuscript is interesting, but I argue about the clinical meaning of the results as you did not compare to controls. Please, see my comments below”:
- Reply 0: Thanks for your considerations and excellent review process in order to improve the quality of this manuscript. We agree with this reviewer and we have added a specific limitation for future projects using case and control studies.
- Changes in the text: Discussion; 4.1. Limitations Page 10; Lines 377-382: “Using the secondary analysis of the baseline sample data of a randomized clinical trial (NCT04582812), the present cross-sectional study tried to describe the respiratory and clinical differences between female and male patients suffering from non-specific LBP, although authors recognized that the comparison of cases and control participants matched-paired by gender could improve the specific comparisons among male and female participants with and without non-specific LBP [12]”.
Comment 1: “- Posteriori means post hoc sample size calculation? If yes, please use the actual description.”
- Reply 1.0: Thanks for your considerations. We apologize for the inconvenience, and we have removed the term “posteriori” to avoid confusion with to post-hoc sample size calculation. In addition, the explanation has been accurately detailed in a correct way.
- Changes in the text: Methods; 2.2. Sample size calculation; Page 3; Lines 123-126: “A sample size calculation was performed to justify the sample required obtained from the baseline data of a randomized clinical trial (NCT04582812) by the difference between 2 independent groups of the version 3.1.9.2 of the G*Power program (G*Power©, University of Dusseldorf; Germany) [34]….”.
Comment 2: “- If you are performing a post hoc sample size analysis, in fact, you are performing a sample power analysis. You already have the effect size, obtained from your actual sample. Please, clarify and change in your current draft.”:
- Reply 2: Thanks for your considerations. Again, sorry for the inconvenience and misunderstanding. We have maintained the initial sample size calculation due to our study was not based on a post hoc sample size analysis, and we have used a general sample size calculation according to our initial sample size description
- Changes in the text: Methods; 2.2. Sample size calculation; Page 3; Lines 123-129: “A sample size calculation was performed to justify the sample required obtained from the baseline data of a randomized clinical trial (NCT04582812) by the difference between 2 independent groups of the version 3.1.9.2 of the G*Power program (G*Power©, University of Dusseldorf; Germany) [34]. Using a general large effect size of Cohen d = 0.80 [28], a two-tailed hypothesis, an error probability of α = 0.05, a power of 1-β probability error = 0.80 and a randomization ratio of N2/N1 = 1, a sample size of 52 participants was necessary, divided into 2 groups of 26 women and 26 men”.
Comment 3: “- "Inclusion criteria comprised patients with a medical diagnosis of bilateral 110 non-specific LBP" - How those patients were classified as non-specifi LBP? Were they assessed using a questionnaire (Start-Back, Oswestry, Rolland Moris, etc)? Did you confirm the diagnostic? How the mechanic cause was excluded?”:
- Reply 3: Thanks for your useful commentaries. We followed the non-specific LBP criteria of prior publications and we have clarified the required information and some limitations have been added according to your suggestions.
- Changes in the text: Methods; 2.2. Study sample; Page 3; Lines 133-144: “Inclusion criteria comprised patients with a prior medical diagnosis of bilateral non-specific LBP for more than 6 weeks considering the main pain location referred by the patient between the bi-iliac and subcostal lines with a bilateral positive active straight leg raise test and aged between 18 and 65 years old [12]. The exclusion criteria comprised patients taking painkillers medication, who has received physiotherapy programs in the last 6 months, with a prior medical diagnosis of congenital lumbar disorders, rheumatic or neuromuscular disorders, body mass index (BMI) greater than 31 kg/m2, previous diagnosis of respiratory or neurological pathology, previous surgery and lower limb pathology (including fractures, sprains or joint instability), skin disorders, inability to follow instructions during the study, pregnancy, and the presence of hyperventilation syndrome assessed by the Nijmegen test greater than 24 points, skin disorders, inability to follow instructions during the study or pregnancy [12,32,36]”.
- Changes in the text: Discussion; 4.2. Limitations; Page 10; Lines 395-399: “Lastly, our study considered non-specific LBP criteria according to prior study’s methodology [12], but future studies should include questionnaires or medical imaging techniques to confirm this diagnosis in a more accurate manner in addition to include more descriptive information such as social or working status of the study participants [55]”.
Comment 4: “- Non-parametric data must be presented as median, minimum and maximum values”:
- Reply 4: Thanks for your valuable considerations. All data have been described considering parametric and non-parametric descriptors.
- Changes in the text: Methods; 2.7. Statistical analyses; Page 6: Lines 258-260: “Parametric and non-parametric data were described using the mean ± standard deviation (SD) and upper and lower limits of the 95% C, as well as the median ± interquartile range (IR) and maximum and minimum values range”.
- Changes in the text: Results; 3.1. Descriptive and physical data; Table 1; Page 7: Lines 292-293: “Table 1. Descriptive data for male and female patients with non-specific LBP”.
Descriptive and physical data (n = 90) |
Female LBP patients (n = 45) |
Male LBP patients (n = 45) |
P-value |
|
||
Mean ± SD (95% CI) |
Median ± IR (Range) |
Mean ± SD (95% CI) |
Median ± IR (Range) |
|
||
Age (y) |
49.15 ± 8.89 (46.48 – 51.82) |
50.00 ± 11.00 (24.00 – 62.00) |
45.75 ± 13.05 (41.83 – 49.67) |
50.00 ± 14.50 (18.00 – 64.00) |
.343† |
|
Height (cm) |
162.76 ± 4.73 (161.33 – 164.18) |
163.00 ± 6.00 (152.00 – 172.00) |
177.98 ± 6.92 (175.90 – 180.06) |
178.00 ± 10.00 (162.00 – 195.00) |
<.001† |
|
Weight (kg) |
64.64 ± 9.68 (61.77 – 67.52) |
63.00 ± 17.50 (49.00 – 80.00) |
81.84 ± 11.22 (78.47 – 85.21) |
82.00 ± 16.50 (49.00 – 105.00) |
<.001* |
|
BMI (kg / m2) |
24.37 ± 3.28 (23.38 – 25.36) |
26.61 ± 5.33 (19.38 – 30.48) |
25.80± 2.95 (24.91 – 26.68) |
26.17 ± 3.61 (16.76 – 30.67) |
.033* |
|
IPAQ (METs / min / week) |
1619.34 ± 1136.88 (1277.78 – 1960.90) |
1404.34 ± 1455.00 (160.00 – 5544.00) |
2416.77 ± 1888.78 (1849.31 – 2984.22) |
1584.00 ± 2385.00 (184.80 – 8586.00) |
.062† |
|
Nijmegen (scores) |
13.95 ± 5.57 (12.28 – 15.62) |
14.00 ± 7.50 (0.00 – 22.00) |
11.02 ± 6.20 (9.15 – 12.88) |
11.00 ± 10.00 (2.00 – 23.00) |
.020* |
|
Abbreviations: BMI, body mass index; CI, confidence interval; IPAQ, International Physical Activity Questionnaire; IR, interquartile range; LBP, low back pain; METs/min/week, metabolic equivalent index per minute per week; P < .05 was significant (in bold) considering a 95% CI. *Student t - test for independent samples was analyzed. †Mann - Whitney U test for independent samples was analyzed. |
- Changes in the text: Results; 3.2. Respiratory outcomes; Table 2; Page 8 Lines 304-305: “Table 2. Comparisons for respiratory outcomes between female and male patients with non-specific LBP”.
Respiratory outcome differences (n = 90) |
Female LBP patients (n = 45) |
Male LBP patients (n = 45) |
Cohen d |
P-value |
|
|||
Mean ± SD (95% CI) |
Median ± IR (Range) |
Mean ± SD (95% CI) |
Median ± IR (Range) |
|
||||
Right diaphragm thickness at Tins (cm) |
0.20 ± 0.06 (0.18 – 0.22) |
0.19 ± 0.00 (0.00 – 0.36) |
0.22 ± 0.06 (0.20 – 0.24) |
0.22 ± 0.10 (0.12 – 0.38) |
0.33 |
.197* |
|
|
Right diaphragm thickness at Texp (cm) |
0.18 ± 0.04 (0.16 – 0.19) |
0.17 ± 0.05 (0.11 – 0.35) |
0.19 ± 0.05 (0.17 – 0.21) |
0.18 ± 0.07 (0.10 – 0.34) |
0.22 |
.253† |
|
|
Right diaphragm thickness at Tins-exp (cm) |
0.02± 0.05 (0.00 – 0.04) |
0.02 ± 0.04 (-0.17 – 0.18) |
0.02 ± 0.03 (0.01 – 0.03) |
0.02 ± 0.04 (-0.03 – 0.12) |
0.00 |
.651† |
|
|
Left diaphragm thickness at Tins (cm) |
0.23 ± 0.07 (0.20 – 0.25) |
0.21 ± 0.08 (0.11 – 0.48) |
0.21 ± 0.05 (0.19 – 0.22) |
0.21 ± 0.07 (0.12 – 0.38) |
0.32 |
.264† |
|
|
Left diaphragm thickness (cm) at Texp |
0.19 ± 0.05 (0.19 – 0.21) |
0.19 ± 0.09 (0.10 – 0.33) |
0.18 ± 0.05 (0.16 – 0.19) |
0.17 ± 0.06 (0.08 – 0.31) |
0.20 |
.285* |
|
|
Left diaphragm thickness at Tins-exp (cm) |
0.03 ± 0.04 (0.02 – 0.04) |
0.02 ± 0.05 (-0.05 – 0.16) |
0.02 ± 0.03 (0.01 – 0.03) |
0.02 ± 0.03 (-0.04 – 0.18) |
0.28 |
.990† |
|
|
MIP (cm H2O) |
53.42 ± 19.17 (47.66 – 59.18) |
48.67 ± 25.33 (17.67 – 100.33) |
85.68 ± 30.62 (76.48 – 94.88) |
84.67 ± 46.00 (33.67 – 153.33) |
1.26 |
<.001† |
|
|
MEP (cm H2O) |
76.88 ± 24.79 (69.43 – 84.33) |
77.33 ± 27.50 (27.73 – 150.33) |
127.55 ± 37.77 (116.20 – 138.90) |
128.33 ± 54.17 (52.67 – 185.67) |
1.58 |
<.001† |
|
|
FEV1 (L) |
2.51 ± 0.56 (2.34 – 2.68) |
2.51 ± 0.64 (1.37 – 3.69) |
3.44 ± 0.69 (3.23 – 3.64) |
3.57 ± 0.88 (1.87 – 4.93) |
1.48 |
<.001* |
|
|
FVC (L) |
2.68 ± 0.63 (2.49 – 2.87) |
2.72 ± 0.85 (1.39 – 3.96) |
3.69 ± 0.84 (3.43 – 3.94) |
3.80 ± 1.07 (1.87 – 6.03) |
1.36 |
<.001* |
|
|
FEV1 / FVC (%) |
93.70 ± 5.70 (91.99 – 95.41) |
95.47 ± 9.41 (78.90 – 99.83) |
93.95 ± 6.59 (91.97 – 95.93) |
96.58 ± 9.80 (75.80 – 99.94) |
0.04 |
.264† |
|
|
Abbreviations: CI, confidence interval; FEV1, forced expiratory volume for 1 second; FVC, forced vital capacity; IR, interquartile range; LBP, low back pain; MEP, maximum expiratory pressure; MIP, maximum inspiratory pressure; Tins, maximum inspiration time; Texp, maximum expiration time. P < .05 was significant considering a 95% CI (in bold). *Student t-test for independent samples was applied. †Mann-Whitney U test for independent samples was applied. |
||||||||
- Changes in the text: Results; 3.3. Clinical outcomes; Table 3; Pages 8-9: Lines 317-319: “Table 3. Comparisons for clinical outcomes between female and male patients with non-specific LBP”.
Clinical outcome differences (n = 90) |
Female LBP patients (n = 45) |
Male LBP patients (n = 45) |
Cohen d |
P-value |
|
|||
Mean ± SD (95% CI) |
Median ± IR (Range) |
Mean ± SD (95% CI) |
Median ± IR (Range) |
|
||||
VAS (scores) |
4.91 ± 1.96 (4.32 – 5.50) |
4.80 ± 3.10 (1.50 – 8.90) |
4.82 ± 1.74 (4.29 – 5.34) |
5.00 ± 2.40 (0.70 – 8.80) |
0.04 |
.808* |
|
|
Paraspinal right PPT (kg / cm2) |
3.72 ± 1.35 (3.31 – 4.13) |
3.80 ± 2.20 (1.40 – 6.33) |
5.53 ± 1.84 (4.98 – 6.08) |
5.67 ± 2.65 (1.90 – 10.00) |
1.12 |
<.001* |
|
|
Paraspinal left PPT (kg / cm2) |
3.70 ± 1.29 (3.31 – 4.09) |
3.67 ± 1.84 (1.30 – 6.80) |
5.48 ± 2.00 (4.87 – 6.08) |
5.47 ± 2.74 (1.50 – 9.93) |
1.05 |
<.001† |
|
|
RMDQ (scores) |
5.06 ± 3.15 (4.11 – 6.01) |
4.00 ± 3.50 (1.00 – 12.00) |
3.51 ± 2.77 (2.67 – 3.34) |
3.00 ± 2.50 (0.00 – 11.00) |
0.52 |
.009† |
|
|
SF - 12 Physical health (optimal normalized values) |
62.55 ± 18.06 (57.12 – 67.98) |
64.00 ± 25.50 (0.00 – 86.00) |
70.80 ± 14.12 (66.55 – 75.04) |
71.00 ± 18.50 (36.00 – 93.00) |
0.50 |
.023† |
|
|
SF - 12 Mental health (optimal normalized values) |
64.20 ± 15.57 (59.52 – 68.87) |
67.00 ± 19.00 (29.00 – 95.00) |
67.86 ± 12.02 (64.25 – 71.47) |
67.00 ± 14.00 (38.00 – 90.00) |
0.26 |
.280† |
|
|
SF - 12 Total score (optimal normalized values) |
63.48 ± 15.31 (58.88 – 68.09) |
66.00 ± 20.00 (17.00 – 91.00) |
69.02 ± 9.71 (66.10 – 71.94) |
71.00 ± 12.50 (43.00 – 89.00) |
0.43 |
.064† |
|
|
Abbreviations: CI, confidence interval; IR, interquartile range; LBP, low back pain; MEP, maximum expiratory pressures; MIP, maximum inspiratory pressures; PPT, pressure pain threshold; RMDQ, Roland-Morris Disability Questionnaire; SF - 12, Short-Form 12-items health questionnaire; VAS, visual analog scale. P < .05 was significant considering a 95% CI (in bold). *Student t-test for independent samples was applied. †Mann-Whitney U test for independent samples was applied. |
||||||||
Comment 5: “- Please, explain the clinical value of comparing male vs. female, as the anthropometrics are distinct. Can those characteristics act as co-variables? If yes, their interference should be accounted for your results?”:
- Reply 5: Thanks for your useful considerations. According to your recommendations, we have performed multivariate regression analyses using linear regression models in order to detail the influence of the descriptive and physical data and gender on the respiratory outcomes that showed statistically significant differences between both study groups. We have included new statistical data and reference, novel regression analyses and these findings were added in the conclusion of this study. Thanks for your fine review.
Changes in the text: 2. Methods; 2.7. Statistical analyses; Pages 6-7: Lines 268-279: “Lastly, the influence of the descriptive and physical data and gender on the respiratory outcomes that presented statistically significant differences between female and male patients with non-specific LBP was predicted by multivariate linear regression analyses. Four linear regression analyses were carried out for each statistically significant respiratory outcome using the “stepwise selection” method. Each regression coefficient value (R2) was calculated to establish the adjustment quality according to the required number of participants per variable [51]. Therefore, age (years), height (cm), weight (kg), BMI (kg/cm2), IPAQ and Nijmegen scores and gender (females = 1; males = 2) were included as independent variables. Each aforementioned statistically significant respiratory outcome measurement was included as dependent variable. Indeed, the pre-established F probabilities were between Pin of .05 and Pout of .10, respectively [12,23]”.
- Austin, P. C.; Steyerberg, E. W. The number of subjects per variable required in linear regression analyses. J. Clin. Epidemiol. 2015, 68, 627–636, doi:10.1016/j.jclinepi.2014.12.014.
Changes in the text: 3. Results; 3.4. Multivariate linear regression analyses; Page 9: Lines 319-330: “The first linear regression model (R2 = 0.290) predicted higher MIP values based on the male gender (R2 = 0.290; β = 32.226; F(1,88) = 35.893; P Ë‚ .001). Next, the second linear regression model (R2 = 0.391) predicted greater MEP values based on the male gender (R2 = 0.391; β = 50.667; F(1,88) = 56.572; P Ë‚ 0001). Also, the third linear regression model (R2 = 0.351) predicted higher FEV1 based on the male gender (R2 = 0.351; β = 0.924; F(1,88) = 47.624; P Ë‚ .001). Finally, the last linear regression model (R2 = 0.355) predicted greater FVC values based on the male gender (R2 = 0.351; β = 0.924; F(1,88) = 47.624; P Ë‚ .001) and higher IPAQ scores (R2 = 0.039; β = 1.098; F(1,87) = 5.295; P = .024). The rest of independent variables, such as height, weight, BMI, and Nijmegen scores were excluded from the prediction models according to the pre-established F probabilities for Pin of .05 and Pout of .10, and thus did not predict or influence these respiratory outcomes”.
Changes in the text: 5. Conclusions; Page 11: Lines 420-424: “Gender – based respiratory differences were presented for maximum respiratory pressures and pulmonary function in patients with non-specific LBP. Women presented greater inspiratory and expiratory muscle weakness as well as worse lung function, although these differences were not linked to diaphragm thickness during normal breathing. These worse respiratory outcomes were mainly predicted by female gender…”
Comment 6: “- The fundamental question about the results is: Compared to healthy controls, would those patients show any marked differences? Why didn't you compare the results to controls?”:
- Reply 6: Thanks for your valuable considerations. We agree with this reviewer that the comparisons of case control studies may add value to the present study. Nevertheless, this secondary analysis was obtained from the baseline sample data of a randomized clinical trial including only patients who suffered from non-specific LBP, not controls, and thus it was initially designed as a cross-sectional study. We have included these explanations in the limitations section of our study. Thanks for your considerations to improve our study.
- Changes in the text: Discussion; 4.1. Limitations Page 10; Lines 377-382: “Using the secondary analysis of the baseline sample data of a randomized clinical trial (NCT04582812), the present cross-sectional study tried to describe the respiratory and clinical differences between female and male patients suffering from non-specific LBP, although authors recognized that the comparison of cases and control participants matched-paired by gender could improve the specific comparisons among male and female participants with and without non-specific LBP [12]”.
Comment 7: “- The discussion is very superficial. The authors must put more effort to explain the results, contrast them to the literature, show and discuss the clinical significance of the current findings, and report the highlights of the present study (take-home messages)”:
- Reply 7: Thanks for your useful considerations. The discussion has been expanded including novel data and references. The limitations section has been placed before future recommendations for a better understanding. The future recommendations section has included take-home messages and clinical recommendations. Again, thanks for the improvement of our manuscript.
Changes in the text: 4. Discussion; Page 10: Lines 355-366: “Mainly, female gender predicted lower MIP, MEP, FEV1 and FVC values in patients who suffered from non-specific LBP according to our multivariate linear regression models (R2 = 0.290 – 0.391). Previously, men showed an increased diaphragm thickness change compared with women healthy adults, which was positively correlated with lung function parameters and respiratory muscle strength [58]. Our findings were in accordance with spirometry parameters and respiratory pressures differences by gender and in contrast with the differences reported in diaphragm thickness in healthy participants [25,59], suggesting the absence of diaphragm thickness differences by gender in patients with LBP. Possibly, the inhibition of the diaphragm function presented in patients with non-specific LBP suggesting an altered breathing pattern during lumbopelvic motor control may explain the dilution of the diaphragm thickness differences between men and women under this condition [60].”.
- Lomauro, A.; Aliverti, A. Sex differences in respiratory function. Breathe 2018, 14, 131, doi:10.1183/20734735.000318.
- Oguri, M.; Okanishi, T.; Ikeguchi, T.; Ogo, K.; Kanai, S.; Maegaki, Y.; Wada, S.; Himoto, T. Influence of gender on diaphragm thickness using a method for determining intima media thickness in healthy young adults. BMC Med. Imaging 2022, 22, doi:10.1186/S12880-022-00748-Y.
- Zakaria, R.; Harif, N.; Al-Rahbi, B.; Aziz, C. B. A.; Ahmad, A. H. Gender Differences and Obesity Influence on Pulmonary Function Parameters. Oman Med. J. 2019, 34, 44, doi:10.5001/OMJ.2019.07.
- Roussel, N.; Nijs, J.; Truijen, S.; Vervecken, L.; Mottram, S.; Stassijns, G. Altered breathing patterns during lumbopelvic motor control tests in chronic low back pain: a case–control study. Eur. Spine J. 2009, 18, 1066, doi:10.1007/S00586-009-1020-Y
Changes in the text: 4. Discussion; 4.1. Limitations; Page 10: Lines 377-382: “Using the secondary analysis of the baseline sample data of a randomized clinical trial (NCT04582812), the present cross-sectional study tried to describe the respiratory and clinical differences between female and male patients suffering from non-specific LBP, although authors recognized that the comparison of cases and control participants matched-paired by gender could improve the specific comparisons among male and female participants with and without non-specific LBP [12]”.
- Changes in the text: Discussion; 4.1. Limitations; Page 10; Lines 396-399: “Lastly, our study considered non-specific LBP criteria according to prior study’s methodology [12], but future studies should include questionnaires or medical imaging techniques to confirm this diagnosis in a more accurate manner in addition to include more descriptive information such as social or working status of the study participants [55]”.
Changes in the text: 4. Discussion; 4.2. Future recommendations; Page 11: Lines 407-417: “As clinical recommendations and take-home messages, rehabilitation protocols should pay especially attention to improve diaphragm strength, lung function and clinical outcomes in women with non-specific LBP according to our current findings suggestion worse results compared with men in line with prior studies in healthy adults [25,58,59]. Next, the motor control diaphragm reeducation during normal breathing should be applied with the same emphasis in both women and men with LBP in line with our study finding reporting no diaphragm thickness differences by gender and the altered motor control reported in patients suffering from this condition [60]. Finally, study only described baseline data differences by gender, but clinical trials should determine the influence of gender on the respiratory and clinical outcomes during breathing reeducation interventions in patients with non-specific LBP [22,23].”.
- Lomauro, A.; Aliverti, A. Sex differences in respiratory function. Breathe 2018, 14, 131, doi:10.1183/20734735.000318
- Oguri, M.; Okanishi, T.; Ikeguchi, T.; Ogo, K.; Kanai, S.; Maegaki, Y.; Wada, S.; Himoto, T. Influence of gender on diaphragm thickness using a method for determining intima media thickness in healthy young adults. BMC Med. Imaging 2022, 22, doi:10.1186/S12880-022-00748-Y.
- Zakaria, R.; Harif, N.; Al-Rahbi, B.; Aziz, C. B. A.; Ahmad, A. H. Gender Differences and Obesity Influence on Pulmonary Function Parameters. Oman Med. J. 2019, 34, 44, doi:10.5001/OMJ.2019.07.
- Roussel, N.; Nijs, J.; Truijen, S.; Vervecken, L.; Mottram, S.; Stassijns, G. Altered breathing patterns during lumbopelvic motor control tests in chronic low back pain: a case–control study. Eur. Spine J. 2009, 18, 1066, doi:10.1007/S00586-009-1020-Y.
Comment 8: “- I strongly recommend to deposit the raw data in a open repository”:
- Reply 8: Thanks for your valuable recommendations. According to our initial proposal project, raw data will be available as a supplementary file with the future clinical trial publication (NCT04582812) due to this study is a secondary analysis of the baseline data of this clinical trial, as we initially detailed in the methods section of this manuscript. Thus, we have included this information in the manuscript to detail the availability of the raw data. Thanks for all.
Changes in the text: Data Availability Statement; Page 12: Lines 467-469: “Raw data will be available as a supplementary file with the future clinical trial publication (NCT04582812) due to this study is a secondary analysis of the baseline data of this clinical trial”.
Comment 10: “Please, check the entire text. Some typos were detected.”:
- Reply 10: Thanks for your great effort and valuable recommendations. The manuscript has been entirely checked to correct typos.
Thanks for your valuable commentaries which have permitted us to improve the quality of the manuscript.
Reviewer 2 Report
Comments and Suggestions for Authors
This manuscript examines the gender respiratory differences of bilateral diaphragm thickness, respiratory pressures and pulmonary function in patients with low back pain. I have with interest read your manuscript. I do have a number of concerns about the manuscript in its current form. I have summarized all the issues in the sections in the attached file. These comments are intended in the light of constructive feedback.

Minor problems on the quality of english language
Author Response
DATE: 05/20/2024
To: Mr. Chauncey Jiang
Section Managing Editor
Tomography
ID: Manuscript ID: tomography-2999292; Title: “A Secondary Analysis of Gender Respiratory Features for Ultrasonography Bilateral Diaphragm Thickness, Respiratory Pressures and Pulmonary Function in Low Back Pain”
Dear Mr. Chauncey Jiang
Section Managing Editor
Tomography:
We appreciate the opportunity to revise our R1 manuscript ID tomography-2999292, updated and entitled "A Secondary Analysis of Gender Respiratory Features for Ultrasonography Bilateral Diaphragm Thickness, Respiratory Pressures and Pulmonary Function in Low Back Pain” to be considered for publication in the original article manuscript category of the Special Issue “Novel Imaging Advances in Physiotherapy” of Tomography. We trust that you will find the current version informative to your readership and acceptable for publication.
Thanks for your review commentaries in order to improve the quality of the manuscript. A deep and substantial modification has been carried out according to your suggestions.
Modifications to the manuscript text are denoted by line numbers and yellow highlighted in the marked-up copy of our manuscript. Please find our responses to each reviewers and editor comment below.
Sincerely, the authors
RESPONSES TO EDITOR'S COMMENTS:
Thanks for your commentaries in order to improve the quality of the manuscript. A deep and substantial modification has been carried out according to your suggestions.
- Comments
“Your manuscript has been reviewed by experts in the field and we request that you make major revisions before it is processed further. Please revise your manuscript according to the reviewers' comments and upload the revised file within 10 days. Please click on the "Peer Review Reports" below to find the reviewers' comments and the version of your manuscript to be used for your revisions”:
- Thanks for your considerations and review process in order to improve the quality of this manuscript. Please, see complete review in response to each reviewer´s suggestion. Thanks.
RESPONSES TO REVIEWERS' COMMENTS:
Reviewer 2:
Thanks for your commentaries in order to improve the quality of the manuscript. A deep and substantial modification has been carried out according to your suggestions.
General comments
Comment 0: “This manuscript examines the gender respiratory differences of bilateral diaphragm thickness, respiratory pressures and pulmonary function in patients with low back pain. I have with interest read your manuscript. I do have a number of concerns about the manuscript in its current form. I have summarized all the issues in the sections below. These comments are intended in the light of constructive feedback”:
- Reply 0: Thanks for your considerations and excellent review process in order to improve the quality of this manuscript. Please, see below the required corrections.
Abstract
Comment 1: “1. Line 19. Please add “the aim of the present study””:
- Reply 1: Thanks for your considerations. The required sentence of the abstract has been modified following your suggestions.
- Changes in the text: Abstract; Page 1; Lines 18-20: “The aim of the present study was to determine the gender respiratory differences of bilateral diaphragm thickness, respiratory pressures and pulmonary function in patients with low back pain (LBP)”.
Introduction
Comment 2: “1. Please explain the term non-specific LBP, what factors causing a non-specific LBP etc.”:
- Reply 2: Thanks for your valuable considerations. The introduction has been expanded explained the definition and factors causing non-specific LBP adding novel references.
- Changes in the text: Introduction; Page 1; Lines 35-39: “Non-specific low back pain (LBP) may be defined as a complex disorder without any specific condition or structural reason in the spine to explain the pain in this body region, including factors from various dimensions, such as movement, pain sensitivity, psychological aspects, and work conditions, which may influence both central and peripheral nociceptive processes [1,2].”
- Rabey, M.; Smith, A.; Kent, P.; Beales, D.; Slater, H.; O’Sullivan, P. Chronic low back pain is highly individualised: patterns of classification across three unidimensional subgrouping analyses. Scand. J. pain 2019, 19, doi:10.1515/SJPAIN-2019-0073.
- Maher, C.; Underwood, M.; Buchbinder, R. Non-specific low back pain. Lancet (London, England) 2017, 389, 736–747, doi:10.1016/S0140-6736(16)30970-9.
Comment 3: “2. Line 48-49. Please make a better association between non-specific LBP and diaphragm dysfunction. You briefly mentioned the relationship of these two variables. Please analyzed more this connection and why is important to examine the rationale of the study”:
- Reply 3: Thanks for your useful considerations. A new paragraph with novel references has been added in the introduction section to justify the connection between non-specific LBP and diaphragm dysfunction according to your suggestions.
- Changes in the text: Introduction; Pages 2; Line 54-58: “Patients suffering from non-specific LBP presented respiratory alterations [13] which appeared to be associated with a thinner diaphragm muscle [12]. Therefore, respiratory muscles weakness and worse lung function in conjunction with the abnormal position of the diaphragm dome and the altered postural motor control were also shown in these patients [14–16].”
- Calvo-Lobo, C.; Almazán-Polo, J.; Becerro-de-Bengoa-Vallejo, R.; Losa-Iglesias, M. E.; Palomo-López, P.; Rodríguez-Sanz, D.; López-López, D. Ultrasonography comparison of diaphragm thickness and excursion between athletes with and without lumbopelvic pain. Phys. Ther. Sport 2019, 37, 128–137, doi:10.1016/j.ptsp.2019.03.015.
- Beeckmans, N.; Vermeersch, A.; Lysens, R.; Van Wambeke, P.; Goossens, N.; Thys, T.; Brumagne, S.; Janssens, L. The presence of respiratory disorders in individuals with low back pain: A systematic review. Man. Ther. 2016, 26, 77–86, doi:10.1016/J.MATH.2016.07.011.
- Uddin, B.; Vaish, H. Evaluation of pulmonary function in patients of non-specific low back pain. Rev. Pesqui. em Fisioter. 2023, 13, e5364, doi:10.17267/2238-2704rpf.2023.e5364.
- Kolar, P.; Sulc, J.; Kyncl, M.; Sanda, J.; Cakrt, O.; Andel, R.; Kumagai, K.; Kobesova, A. Postural function of the diaphragm in persons with and without chronic low back pain. J. Orthop. Sports Phys. Ther. 2012, 42, 352–362, doi:10.2519/jospt.2012.3830.
- Mohan MPT, V.; Paungmali, A.; Sitilerpisan, P.; Hashim, U. F.; Mazlan BPT, M. B.; Nasuha BPT, T. N.; Selangor, M.; Puncak Alam, B.; Alam, P.; MARA Selangor, T. Respiratory characteristics of individuals with non-specific low back pain: A cross-sectional study. Nurs. Health Sci. 2018, 20, 224–230, doi:10.1111/NHS.12406.
Comment 4: “3. Line 57. Please explain “Why gender differences of respiratory features of patients with LBP need to be further studied?”:
- Reply 4: Thanks for your useful considerations. The introduction has been expanded adding new references to explain why gender differences of respiratory features of patients with LBP need to be further studied following your recommendations.
- Changes in the text: Introduction; Page 2; Lines 65-72: “Nevertheless, gender differences of respiratory features of patients with LBP need to be further studied. Anatomically, the length of the diaphragm was reported to be 9% shorter in women than men and also this length in the zone of apposition with respect to the rib cage was smaller in women, which was correlated with worse lung function and weaker respiratory muscles [25]. Pathologically, these respiratory features could be exacerbated under non-specific LBP showing greater respiratory muscle weakness, worse lung function and altered diaphragm [14–16].
- Uddin, B.; Vaish, H. Evaluation of pulmonary function in patients of non-specific low back pain. Rev. Pesqui. em Fisioter. 2023, 13, e5364, doi:10.17267/2238-2704rpf.2023.e5364.
- Kolar, P.; Sulc, J.; Kyncl, M.; Sanda, J.; Cakrt, O.; Andel, R.; Kumagai, K.; Kobesova, A. Postural function of the diaphragm in persons with and without chronic low back pain. J. Orthop. Sports Phys. Ther. 2012, 42, 352–362, doi:10.2519/jospt.2012.3830.
- Mohan MPT, V.; Paungmali, A.; Sitilerpisan, P.; Hashim, U. F.; Mazlan BPT, M. B.; Nasuha BPT, T. N.; Selangor, M.; Puncak Alam, B.; Alam, P.; MARA Selangor, T. Respiratory characteristics of individuals with non-specific low back pain: A cross-sectional study. Nurs. Health Sci. 2018, 20, 224–230, doi:10.1111/NHS.12406.
- Lomauro, A.; Aliverti, A. Sex differences in respiratory function. Breathe 2018, 14, 131, doi:10.1183/20734735.000318.
Methods
Study design, recruitment and ethical aspects
Comment 5: “1. Please describe better so as to clarify the recruitment. As far as I understand this study has used a sample from another previous study of yours? Is that correct?”:
- Reply 5: Thanks for your valuable considerations. The required information has been clarified due to this manuscript was a secondary analysis using the baseline data of a randomized clinical trial.
- Changes in the text: Methods; 2.1. Study design, recruitment and ethical aspects; Page 3; Lines 97-103: “The sample was obtained by a consecutive sampling method from the baseline data of a randomized clinical trial (NCT04582812), which used a simple randomization method to recruit all participants matched paired by sex in the Complutense University of Madrid”.
Comment 6: “2. Line 82. Please clarify what the simple randomization method was?”:
- Reply 6: Thanks for your valuable considerations. In addition to the aforementioned information, we have clarified that a consecutive sampling method was used according to our cross-sectional study (from the baseline sample of the randomized clinical trial, which used a simple randomization process matched paired by sex.
- Changes in the text: Methods; 2.1. Study design, recruitment and ethical aspects; Page 3; Lines 97-103: “The sample was obtained by a consecutive sampling method from the baseline data of a randomized clinical trial (NCT04582812), which used a simple randomization method to recruit all participants matched paired by sex in the Complutense University of Madrid”.
Comment 7: “3. Line 87-89. Please describe in more detail bilateral thoracic orthosis device”:
- Reply 7: Thanks for your review. The bilateral orthosis device has been described according to your suggestions.
- Changes in the text: Methods; 2.1. Study design, recruitment and ethical aspects; Page 3; Lines 104-108: “This device was composed of a thoracic orthosis including 2 bilateral bivalve adapters for 2 holding devices to fix 2 ultrasound probes allowing total thoracic mobility and including two spaces to incorporate ultrasound gel permitting the complete visualization of both right and left last intercostal spaces”.
Study sample
Comment 8: “1. Please add more inclusion criteria. Why have you chosen only 1-2 criteria? For example please add if they have taken painkillers, if they have done physiotherapy program-if yes for how long, duration etc., if they have done a MRI, how it was diagnosed it, functional status problems, going to work or not due to LBP etc. Thus please regarding the inclusion criteria adds more demographical and clinical characteristics of the LBP”:
- Reply 8: Thanks for your useful commentaries. We followed the non-specific LBP criteria of prior publications and we have clarified the required information and some limitations have been added according to your suggestions.
- Changes in the text: Methods; 2.2. Study sample; Page 3; Lines 133-144: “Inclusion criteria comprised patients with a prior medical diagnosis of bilateral non-specific LBP for more than 6 weeks considering the main pain location referred by the patient between the bi-iliac and subcostal lines with a bilateral positive active straight leg raise test and aged between 18 and 65 years old [12]. The exclusion criteria comprised patients taking painkillers medication, who has received physiotherapy programs in the last 6 months, with a prior medical diagnosis of congenital lumbar disorders, rheumatic or neuromuscular disorders, body mass index (BMI) greater than 31 kg/m2, previous diagnosis of respiratory or neurological pathology, previous surgery and lower limb pathology (including fractures, sprains or joint instability), skin disorders, inability to follow instructions during the study, pregnancy, and the presence of hyperventilation syndrome assessed by the Nijmegen test greater than 24 points, skin disorders, inability to follow instructions during the study or pregnancy [12,32,36]”.
- Changes in the text: Discussion; 4.1. Limitations; Page 10; Lines 395-399: “Lastly, our study considered non-specific LBP criteria according to prior study’s methodology [12], but future studies should include questionnaires or medical imaging techniques to confirm this diagnosis in a more accurate manner in addition to include more descriptive information such as social or working status of the study participants [55]”.
Respiratory outcomes
Comment 9: “1. Line 151. Please report why 3 repeated measures and not 4 or 2? Please add references”:
- Reply 9: Thanks for your valuable recommendations. This procedure has been justified and cited in an accurate manner according to your considerations.
Changes in the text: 2. Methods; 2.5. Respiratory outcomes; 2.5.1. Bilateral diaphragm thickness; Page 4: Lines 176-178: “A total of 3 repeated measures were carried out to measure both right and left hemi - diaphragms thickness at Tins, Texp and Tins-exp, using 3 images for each parameter, which has shown the reduction of errors of measurement [32]”.
Clinical outcomes
Comment 10: “1. 1. Please clarify why you have measured pain intensity at rest? Why last week? What criteria have you used to decide that?”:
- Reply 10: Thanks for your valuable recommendations. This measurement has been justified and cited in an accurate manner according to your considerations. In addition, some limitations have been acknowledged regarding this issue.
Changes in the text: 2. Methods; 2.6. Clinical outcomes; 2.6.1. Pain intensity; Pages 5-6: Lines 222-228: “Pain intensity measurements were determined using the Visual Analog Scale (VAS) by the self - reported pain intensity average during the last week at rest according to the methodology used in prior similar studies [12,23,32]. This scale was composed of a 100 - mm horizontal line where participants marked their pain intensity since “no pain” at the left side to the “worst pain imaginable” at the right side. This scale presented as adequate reliability as validity presenting an ICC of 0.88 to determine pain intensity in the last week with a correlation of r = 0.74 with other scales of pain intensity [5,44]”.
Changes in the text: 4. Discussion; 4.1. Limitations; Page 10: Lines 388-393: “Pain intensity was determining indicating that patients self-reported their LBP intensity average during the last week at rest. Authors acknowledge that the moderate pain intensity mean presented in both women and men could have been confound by pain intensity during daily life activities with respect to pain intensity at rest and this issue should be considered in future studies differentiating between pain intensity at rest and during physical activity [5,44]”.
Comment 11: “2. Please clarify who make all the clinical assessments of the sample. Was the same examiner?”:
- Reply 11: Thanks for your valuable recommendations. The required information has been clarified according to your suggestions.
Changes in the text: 2. Methods; 2.1. Study design, recruitment and ethical aspects; Page 3: Lines 119-121: “Descriptive data and outcomes were measured by a physician together with other experienced evaluators using self-reported and inter-examiner reliable tools, applying a blinded evaluation for ultrasound images by numerical coding [23]”.
.
Comment 12: “3. Please clarify why have you used the Roland Morris Disability Q and not any other?”:
- Reply 12: Thanks for your valuable recommendations. The required information has been also clarified according to your suggestions.
Changes in the text: 2. Methods; 2.6. Clinical outcomes; 2.6.3. Disability; Page 6: Lines 240-242: “This clinical outcome was measured using the Spanish Roland-Morris Disability Questionnaire (RMDQ), which has widely used to self-report the disability linked to LBP following prior similar studies [12,23,32]…”.
Discussion
Comment 14: “1. The discussion seems a small one. Please add more references in order to make a more in depth discussion”:
- Reply 14: Thanks for your useful considerations. The discussion has been expanded including novel data and references. The limitations section has been placed before future recommendations for a better understanding. The future recommendations section has included take-home messages and clinical recommendations. Again, thanks for the improvement of our manuscript.
Changes in the text: 4. Discussion; Page 9: Lines 355-366: “Mainly, female gender predicted lower MIP, MEP, FEV1 and FVC values in patients who suffered from non-specific LBP according to our multivariate linear regression models (R2 = 0.290 – 0.391). Previously, men showed an increased diaphragm thickness change compared with women healthy adults, which was positively correlated with lung function parameters and respiratory muscle strength [58]. Our findings were in accordance with spirometry parameters and respiratory pressures differences by gender and in contrast with the differences reported in diaphragm thickness in healthy participants [25,59], suggesting the absence of diaphragm thickness differences by gender in patients with LBP. Possibly, the inhibition of the diaphragm function presented in patients with non-specific LBP suggesting an altered breathing pattern during lumbopelvic motor control may explain the dilution of the diaphragm thickness differences between men and women under this condition [60].”.
- Lomauro, A.; Aliverti, A. Sex differences in respiratory function. Breathe 2018, 14, 131, doi:10.1183/20734735.000318.
- Oguri, M.; Okanishi, T.; Ikeguchi, T.; Ogo, K.; Kanai, S.; Maegaki, Y.; Wada, S.; Himoto, T. Influence of gender on diaphragm thickness using a method for determining intima media thickness in healthy young adults. BMC Med. Imaging 2022, 22, doi:10.1186/S12880-022-00748-Y.
- Zakaria, R.; Harif, N.; Al-Rahbi, B.; Aziz, C. B. A.; Ahmad, A. H. Gender Differences and Obesity Influence on Pulmonary Function Parameters. Oman Med. J. 2019, 34, 44, doi:10.5001/OMJ.2019.07.
- Roussel, N.; Nijs, J.; Truijen, S.; Vervecken, L.; Mottram, S.; Stassijns, G. Altered breathing patterns during lumbopelvic motor control tests in chronic low back pain: a case–control study. Eur. Spine J. 2009, 18, 1066, doi:10.1007/S00586-009-1020-Y
Changes in the text: 4. Discussion; 4.1. Limitations; Page 10: Lines 377-382: “Using the secondary analysis of the baseline sample data of a randomized clinical trial (NCT04582812), the present cross-sectional study tried to describe the respiratory and clinical differences between female and male patients suffering from non-specific LBP, although authors recognized that the comparison of cases and control participants matched-paired by gender could improve the specific comparisons among male and female participants with and without non-specific LBP [12]”.
- Changes in the text: Discussion; 4.2. Limitations; Page 10; Lines 395-399: “Lastly, our study considered non-specific LBP criteria according to prior study’s methodology [12], but future studies should include questionnaires or medical imaging techniques to confirm this diagnosis in a more accurate manner in addition to include more descriptive information such as social or working status of the study participants [55]”.
Changes in the text: 4. Discussion; 4.2. Future recommendations; Page 11: Lines 407-417: “As clinical recommendations and take-home messages, rehabilitation protocols should pay especially attention to improve diaphragm strength, lung function and clinical outcomes in women with non-specific LBP according to our current findings suggestion worse results compared with men in line with prior studies in healthy adults [25,58,59]. Next, the motor control diaphragm reeducation during normal breathing should be applied with the same emphasis in both women and men with LBP in line with our study finding reporting no diaphragm thickness differences by gender and the altered motor control reported in patients suffering from this condition [60]. Finally, study only described baseline data differences by gender, but clinical trials should determine the influence of gender on the respiratory and clinical outcomes during breathing reeducation interventions in patients with non-specific low back pain [22,23]”.
- Lomauro, A.; Aliverti, A. Sex differences in respiratory function. Breathe 2018, 14, 131, doi:10.1183/20734735.000318.
- Oguri, M.; Okanishi, T.; Ikeguchi, T.; Ogo, K.; Kanai, S.; Maegaki, Y.; Wada, S.; Himoto, T. Influence of gender on diaphragm thickness using a method for determining intima media thickness in healthy young adults. BMC Med. Imaging 2022, 22, doi:10.1186/S12880-022-00748-Y.
- Zakaria, R.; Harif, N.; Al-Rahbi, B.; Aziz, C. B. A.; Ahmad, A. H. Gender Differences and Obesity Influence on Pulmonary Function Parameters. Oman Med. J. 2019, 34, 44, doi:10.5001/OMJ.2019.07.
- Roussel, N.; Nijs, J.; Truijen, S.; Vervecken, L.; Mottram, S.; Stassijns, G. Altered breathing patterns during lumbopelvic motor control tests in chronic low back pain: a case–control study. Eur. Spine J. 2009, 18, 1066, doi:10.1007/S00586-009-1020-Y.
Comment 15: “2. 2. Please provide first the limitations and afterwards the future recommendations”:
- Reply 15: Thanks for your useful considerations. The limitations section has been placed before future recommendations for a better understanding.
- Changes in the text: Discussion; 4.1. Limitations; Page 10: Lines 377-382: “Using the secondary analysis of the baseline sample data of a randomized clinical trial (NCT04582812), the present cross-sectional study tried to describe the respiratory and clinical differences between female and male patients suffering from non-specific LBP, although authors recognized that the comparison of cases and control participants matched-paired by gender could improve the specific comparisons among male and female participants with and without non-specific LBP [12]”.
- Changes in the text: Discussion; 4.1. Limitations; Page 10; Lines 395-399: “Lastly, our study considered non-specific LBP criteria according to prior study’s methodology [12], but future studies should include questionnaires or medical imaging techniques to confirm this diagnosis in a more accurate manner in addition to include more descriptive information such as social or working status of the study participants [55]”.
Changes in the text: 4. Discussion; 4.2. Future recommendations; Page 10: Lines 395-399: “As clinical recommendations and take-home messages, rehabilitation protocols should pay especially attention to improve diaphragm strength, lung function and clinical outcomes in women with non-specific LBP according to our current findings suggestion worse results compared with men in line with prior studies in healthy adults [25,58,59]. Next, the motor control diaphragm reeducation during normal breathing should be applied with the same emphasis in both women and men with LBP in line with our study finding reporting no diaphragm thickness differences by gender and the altered motor control reported in patients suffering from this condition [60]. Finally, study only described baseline data differences by gender, but clinical trials should determine the influence of gender on the respiratory and clinical outcomes during breathing reeducation interventions in patients with non-specific low back pain [22,23]”.
- Lomauro, A.; Aliverti, A. Sex differences in respiratory function. Breathe 2018, 14, 131, doi:10.1183/20734735.000318.
- Oguri, M.; Okanishi, T.; Ikeguchi, T.; Ogo, K.; Kanai, S.; Maegaki, Y.; Wada, S.; Himoto, T. Influence of gender on diaphragm thickness using a method for determining intima media thickness in healthy young adults. BMC Med. Imaging 2022, 22, doi:10.1186/S12880-022-00748-Y.
- Zakaria, R.; Harif, N.; Al-Rahbi, B.; Aziz, C. B. A.; Ahmad, A. H. Gender Differences and Obesity Influence on Pulmonary Function Parameters. Oman Med. J. 2019, 34, 44, doi:10.5001/OMJ.2019.07.
- Roussel, N.; Nijs, J.; Truijen, S.; Vervecken, L.; Mottram, S.; Stassijns, G. Altered breathing patterns during lumbopelvic motor control tests in chronic low back pain: a case–control study. Eur. Spine J. 2009, 18, 1066, doi:10.1007/S00586-009-1020-Y.
Comment 16: “3. Please correct the title of the 4.1. Part as future recommendations”:
- Reply 16: Thanks for your raised considerations. The future recommendations section has been corrected and expanded.
Changes in the text: 4. Discussion; 4.2. Future recommendations; Page 10: Lines 395-399: “As clinical recommendations and take-home messages, rehabilitation protocols should pay especially attention to improve diaphragm strength, lung function and clinical outcomes in women with non-specific LBP according to our current findings suggestion worse results compared with men in line with prior studies in healthy adults [25,58,59]. Next, the motor control diaphragm reeducation during normal breathing should be applied with the same emphasis in both women and men with LBP in line with our study finding reporting no diaphragm thickness differences by gender and the altered motor control reported in patients suffering from this condition [60]. Finally, study only described baseline data differences by gender, but clinical trials should determine the influence of gender on the respiratory and clinical outcomes during breathing reeducation interventions in patients with non-specific low back pain [22,23]”.
- Lomauro, A.; Aliverti, A. Sex differences in respiratory function. Breathe 2018, 14, 131, doi:10.1183/20734735.000318.
- Oguri, M.; Okanishi, T.; Ikeguchi, T.; Ogo, K.; Kanai, S.; Maegaki, Y.; Wada, S.; Himoto, T. Influence of gender on diaphragm thickness using a method for determining intima media thickness in healthy young adults. BMC Med. Imaging 2022, 22, doi:10.1186/S12880-022-00748-Y.
- Zakaria, R.; Harif, N.; Al-Rahbi, B.; Aziz, C. B. A.; Ahmad, A. H. Gender Differences and Obesity Influence on Pulmonary Function Parameters. Oman Med. J. 2019, 34, 44, doi:10.5001/OMJ.2019.07.
- Roussel, N.; Nijs, J.; Truijen, S.; Vervecken, L.; Mottram, S.; Stassijns, G. Altered breathing patterns during lumbopelvic motor control tests in chronic low back pain: a case–control study. Eur. Spine J. 2009, 18, 1066, doi:10.1007/S00586-009-1020-Y.
Comment 17: “4. 4. Please confirm that the content of the 4.1. Part is future recommendations. It seems as it is a conclusion. Please re-write it”:
- Reply 17: Thanks again for your raised considerations. The future recommendations section has included take-home messages and clinical recommendations. Again, thanks for the improvement of our manuscript.
Changes in the text: 4. Discussion; 4.2. Future recommendations; Page 10: Lines 395-399: “As clinical recommendations and take-home messages, rehabilitation protocols should pay especially attention to improve diaphragm strength, lung function and clinical outcomes in women with non-specific LBP according to our current findings suggestion worse results compared with men in line with prior studies in healthy adults [25,58,59]. Next, the motor control diaphragm reeducation during normal breathing should be applied with the same emphasis in both women and men with LBP in line with our study finding reporting no diaphragm thickness differences by gender and the altered motor control reported in patients suffering from this condition [60]. Finally, study only described baseline data differences by gender, but clinical trials should determine the influence of gender on the respiratory and clinical outcomes during breathing reeducation interventions in patients with non-specific low back pain [22,23]”.
- Lomauro, A.; Aliverti, A. Sex differences in respiratory function. Breathe 2018, 14, 131, doi:10.1183/20734735.000318.
- Oguri, M.; Okanishi, T.; Ikeguchi, T.; Ogo, K.; Kanai, S.; Maegaki, Y.; Wada, S.; Himoto, T. Influence of gender on diaphragm thickness using a method for determining intima media thickness in healthy young adults. BMC Med. Imaging 2022, 22, doi:10.1186/S12880-022-00748-Y.
- Zakaria, R.; Harif, N.; Al-Rahbi, B.; Aziz, C. B. A.; Ahmad, A. H. Gender Differences and Obesity Influence on Pulmonary Function Parameters. Oman Med. J. 2019, 34, 44, doi:10.5001/OMJ.2019.07.
- Roussel, N.; Nijs, J.; Truijen, S.; Vervecken, L.; Mottram, S.; Stassijns, G. Altered breathing patterns during lumbopelvic motor control tests in chronic low back pain: a case–control study. Eur. Spine J. 2009, 18, 1066, doi:10.1007/S00586-009-1020-Y.
Comment 18: “Minor problems on the quality of english language.”:
- Reply 18: Thanks for your great effort and valuable recommendations. The manuscript has been entirely checked to revise minor English problems.
Thanks for your valuable commentaries which have permitted us to improve the quality of the manuscript.
Sincerely,
The authors

Round 2
Reviewer 1 Report
Comments and Suggestions for Authors
Dear authors,
Thank you for your effort to improve the manuscript.
The sample size calculation is still not correct considering it is a post hoc analysis. Thus, this should be corrected considering the retrospective characteristics of the study.
Regards.
Author Response
DATE: 05/24/2024
To: Mr. Chauncey Jiang
Section Managing Editor
Tomography
ID: Manuscript ID: tomography-2999292; r2 Minor revisions; Title: “A Secondary Analysis of Gender Respiratory Features for Ultrasonography Bilateral Diaphragm Thickness, Respiratory Pressures and Pulmonary Function in Low Back Pain”
Dear Mr. Chauncey Jiang
Section Managing Editor
Tomography:
We appreciate the opportunity to revise our R2 manuscript ID tomography-2999292 for minor revisions, updated and entitled "A Secondary Analysis of Gender Respiratory Features for Ultrasonography Bilateral Diaphragm Thickness, Respiratory Pressures and Pulmonary Function in Low Back Pain” to be considered for publication in the original article manuscript category of the Special Issue “Novel Imaging Advances in Physiotherapy” of Tomography. We trust that you will find the current version informative to your readership and acceptable for publication.
Thanks for your review commentaries in order to improve the quality of the manuscript. A deep and substantial modification has been carried out according to your suggestions.
Modifications to the manuscript text are denoted by line numbers and yellow highlighted for R1 revisions and blue highlighted for R2 revisions in the marked-up copy of our manuscript. Please find our responses to each reviewer below.
Sincerely, the authors
RESPONSES TO EDITOR'S COMMENTS:
Thanks for your commentaries in order to improve the quality of the manuscript. A deep and substantial modification has been carried out according to your suggestions.
- Comments
“Please revise your manuscript according to the referees’ comments and
upload the revised file within 2 days”:
- Thanks for your considerations and review process in order to improve the quality of this manuscript. Please, see complete review in response to each reviewer´s suggestion. Thanks.
- Modifications to the manuscript text are denoted by line numbers and yellow highlighted for R1 revisions and blue highlighted for R2 revisions in the marked-up copy of our manuscript. Please find our responses to each reviewer below
RESPONSES TO REVIEWERS' COMMENTS:
Reviewer 1:
Thanks for your commentaries in order to improve the quality of the manuscript. A deep and substantial modification has been carried out according to your suggestions.
Comment 1: “Dear authors, thank you for your effort to improve the manuscript. The sample size calculation is still not correct considering it is a post hoc analysis. Thus, this should be corrected considering the retrospective characteristics of the study. Regards.”
- Reply 1: Thanks for your valuable considerations. According to your suggestions, a post hoc sample size calculation has been performed. Again, thanks for the improvement on our manuscript.
- Changes in the text: Methods; 2.2. Sample size calculation; Page 3; Lines 121-126: “A post hoc sample size calculation was performed to justify the achieved power obtained from the baseline data of a randomized clinical trial (NCT04582812) by the difference between 2 independent groups of the version 3.1.9.2 of the G*Power program (G*Power©, University of Dusseldorf; Germany) [34]. Using a general large effect size of Cohen d = 0.80 [35], a two-tailed hypothesis, an error probability of α = 0.05 and the used sample size of 45 women and 45 men, a power (1-β probability error) of 0.96 was obtained”.
Thanks for your valuable commentaries which have permitted us to improve the quality of the manuscript.
Sincerely,
The authors
Reviewer 2 Report
Comments and Suggestions for Authors
The authors have made major changes to the paper and I appreciate their revisions. The manuscript presentation and content has been significantly enhanced. Thank you for their cooperation!
Author Response
DATE: 05/24/2024
To: Mr. Chauncey Jiang
Section Managing Editor
Tomography
ID: Manuscript ID: tomography-2999292; r2 Minor revisions; Title: “A Secondary Analysis of Gender Respiratory Features for Ultrasonography Bilateral Diaphragm Thickness, Respiratory Pressures and Pulmonary Function in Low Back Pain”
Dear Mr. Chauncey Jiang
Section Managing Editor
Tomography:
We appreciate the opportunity to revise our R2 manuscript ID tomography-2999292 for minor revisions, updated and entitled "A Secondary Analysis of Gender Respiratory Features for Ultrasonography Bilateral Diaphragm Thickness, Respiratory Pressures and Pulmonary Function in Low Back Pain” to be considered for publication in the original article manuscript category of the Special Issue “Novel Imaging Advances in Physiotherapy” of Tomography. We trust that you will find the current version informative to your readership and acceptable for publication.
Thanks for your review commentaries in order to improve the quality of the manuscript. A deep and substantial modification has been carried out according to your suggestions.
Modifications to the manuscript text are denoted by line numbers and yellow highlighted for R1 revisions and blue highlighted for R2 revisions in the marked-up copy of our manuscript. Please find our responses to each reviewer below.
Sincerely, the authors
RESPONSES TO EDITOR'S COMMENTS:
Thanks for your commentaries in order to improve the quality of the manuscript. A deep and substantial modification has been carried out according to your suggestions.
- Comments
“Please revise your manuscript according to the referees’ comments and
upload the revised file within 2 days”:
- Thanks for your considerations and review process in order to improve the quality of this manuscript. Please, see complete review in response to each reviewer´s suggestion. Thanks.
- Modifications to the manuscript text are denoted by line numbers and yellow highlighted for R1 revisions and blue highlighted for R2 revisions in the marked-up copy of our manuscript. Please find our responses to each reviewer below
RESPONSES TO REVIEWERS' COMMENTS:
Reviewer 2:
Thanks for your commentaries in order to improve the quality of the manuscript. A deep and substantial modification has been carried out according to your suggestions.
Comment 1: “The authors have made major changes to the paper and I appreciate their revisions. The manuscript presentation and content has been significantly enhanced. Thank you for their cooperation!”:
- Reply 1: Thanks for your considerations and excellent review process in order to improve the quality of this manuscript. It has been a pleasure to respond your raised recommendations.
Thanks for your valuable commentaries which have permitted us to improve the quality of the manuscript.
Sincerely,
The authors